# VIDAR: EMBODIED VIDEO DIFFUSION MODEL FOR GENERALIST MANIPULATION

## ABSTRACT

Scaling general-purpose manipulation to new robot embodiments remains challenging: each platform typically needs large, homogeneous demonstrations, and end-to-end pixel-to-action pipelines may degenerate under background and viewpoint shifts. Based on previous advances in video-based robot control, we present Vidar, consisting of an embodied video diffusion model as the generalizable prior and a masked inverse dynamics model (MIDM) as the adapter. We leverage a video diffusion model pre-trained at Internet scale, and further continuously pre-train it for the embodied domain using 750K multi-view trajectories collected from three real-world robot platforms. For this embodied pre-training, we introduce a unified observation space that jointly encodes robot, camera, task, and scene contexts. The MIDM module learns action-relevant pixel masks without dense labels, grounding the prior into the target embodiment's action space while suppressing distractors. With only ∼20 minutes of human demonstrations on an unseen robot (∼1% of typical data), Vidar outperforms state-of-the-art baselines and generalizes to unseen tasks, backgrounds, and camera layouts. Our results suggest a scalable recipe for "one prior, many embodiments": strong, inexpensive video priors together with minimal on-robot alignment.

## 1 INTRODUCTION

Robotic manipulation spans skills such as stable object handling, in-hand reorientation, and multi-point contact control—capabilities underlying cloth folding, liquid pouring, and tool-assisted assembly. Vision-language-action (VLA) models (O'Neill et al., 2024; Kim et al., 2024; Liu et al., 2024a; Zhang & Yan, 2023) have made encouraging progress via large-scale multimodal pre-training, yet extending them to general-purpose or bimanual settings remains difficult. The core obstacle is control complexity: the action space grows combinatorially with added joints, while success hinges on tight temporal coordination, accurate contact dynamics, and long-horizon reasoning. These factors amplify data requirements and heighten sensitivity to platform-specific details. In practice, progress is constrained by **data scarcity**: human demonstration corpora typically contain only tens to hundreds of hours (Intelligence et al., 2025; Liu et al., 2024a), orders of magnitude smaller than Internet-scale video collections with hundreds of thousands of hours (Wang et al., 2024). Collecting demonstrations is labor-intensive, expensive, and coupled to hardware, leaving a key question: *how can a new robot embodiment achieve precise, generalizable control with limited domain-specific data?*

A natural answer lies in *video*. Unlike text or static images, video is both abundant and intrinsically suited to capture the temporal dynamics and interaction cues—affordances, contacts, motion continuity—that demonstrations aim to convey. Leveraging such signals enables robots to acquire embodiment-agnostic interaction knowledge at scale, and later specialize to new morphologies with far fewer platform-specific samples. To turn this rich modality into a transferable prior, we adopt video generative models, which learn distributions of *plausible, temporally coherent rollouts* rather than task-specific labels. This generative formulation enforces physical consistency, supports counterfactual reasoning about "what could happen next", and shifts dependence from costly demonstrations to abundant raw video. Recent advances in video diffusion models trained on web-scale corpora already show strong semantic grounding and temporal fidelity (Liu et al., 2024b; Wang et al., 2025; Kong et al., 2024; Bao et al., 2024), making them well-suited to serve as general interaction priors for low-shot embodiment alignment. Meanwhile, for robot control, our objective departs from vanilla

video diffusion. Rather than producing photorealistic clips, we require *actionable* rollouts that are consistent with robot actuation, accurate in contact dynamics, and robust across embodiments.

Based on previous advances in using video generative models for robot control (Du et al., 2023), we propose **Video Diffusion for Action Reasoning (Vidar)**, to better explore the transferable prior from videos for efficient bimanual manipulation with decoupled video generation and action prediction. To build a good video prior for embodied control, we propose a three-stage training pipeline: Internet-scale videos are used for general pre-training (where off-the-shelf checkpoints can be adopted), large cross-embodiment robotic datasets are used for embodied domain pre-training, and a small number of robot-specific demonstrations are used for target domain fine-tuning. Specifically for embodied domain pre-training, we align 750K multi-view bimanual clips spanning three robot platforms into a unified observation space. Such pre-training on a unified space ensures control feasibility, promotes physically credible contacts, and mitigates viewpoint and morphology gaps. During inference, we further enhance rollout quality (e.g., physical plausibility, instruction relevance) by applying test-time scaling (Jaech et al., 2024).

On the action side, the main challenge is to decode videos into reliable controls despite background clutter, distractors, and partial observability of hands and tools. We address this with a **Masked Inverse Dynamics Model (MIDM)**, which learns to attend selectively to action-relevant regions without pixel-level supervision. By filtering out irrelevant content, MIDM provides robust action decoding and facilitates the transfer of video priors across domains. Moreover, such a lightweight model can be trained using only a small number of demonstrations. Together, these components transform raw Internet video into a transferable and controllable prior, enabling precise manipulation with only a small number of demonstrations.

Empirically, Vidar achieves state-of-the-art performance on the RoboTwin (Chen et al., 2025) benchmark under the challenging multi-task setting. In the real world, it attains strong performance with only **20 minutes of human demonstrations** (roughly 3 per task) on a previously unseen robotic platform. Despite this minimal supervision, it outperforms leading baselines by large margins—**58%** over VPP (Hu et al., 2024) and **40%** over UniPi (Du et al., 2023). Moreover, it generalizes robustly to novel scenarios, such as environments with reflective surfaces, indicating that video-pretrained priors can support both data-efficient adaptation and semantically grounded control.

## 2 METHOD

In this section, we formally define our problem and present Vidar in detail. The overview of our method is shown in Figure 1.

### 2.1 PROBLEM FORMULATION

This work investigates the challenges of bimanual manipulation for everyday activities, tasks that are inherently resistant to standardization. Our experiments are conducted using the common Aloha robot platform (Liu et al., 2024a; Fu et al., 2024), with detailed hardware specifications given in Appendix F. The problem is formulated as follows.

Let $\mathcal{L}$ denote the language instruction space, $\mathcal{O}$ the observation space, and $\mathcal{A}$ the action space. Our goal is to learn a conditional manipulation policy

$$\pi : \mathcal{L} \times \mathcal{O} \to \mathbb{P}(\mathcal{A}),$$

where $\mathbb{P}(\cdot)$ denotes probability measures over the corresponding space. Learning this policy directly is highly challenging. It requires large-scale demonstrations that jointly cover language, observation, and action, which are expensive and hardware-specific. Moreover, robots differ in sensing, morphology, and viewpoints, making policies learned on one platform difficult to transfer. Finally, successful manipulation depends on fine-grained contact events and long-horizon temporal coherence; photorealistic video generation alone does not guarantee *actionability*.

We address these challenges by elevating the action space to the video domain, where richer semantic information is preserved and an abundance of large-scale data is available for learning a strong, transferable prior. We factorize the policy through the video space $\mathcal{V}$:

$$\pi = I \circ G, \qquad G : \mathcal{L} \times \mathcal{O} \to \mathbb{P}(\mathcal{V}), \quad I : \mathcal{V} \to \mathcal{A}.$$

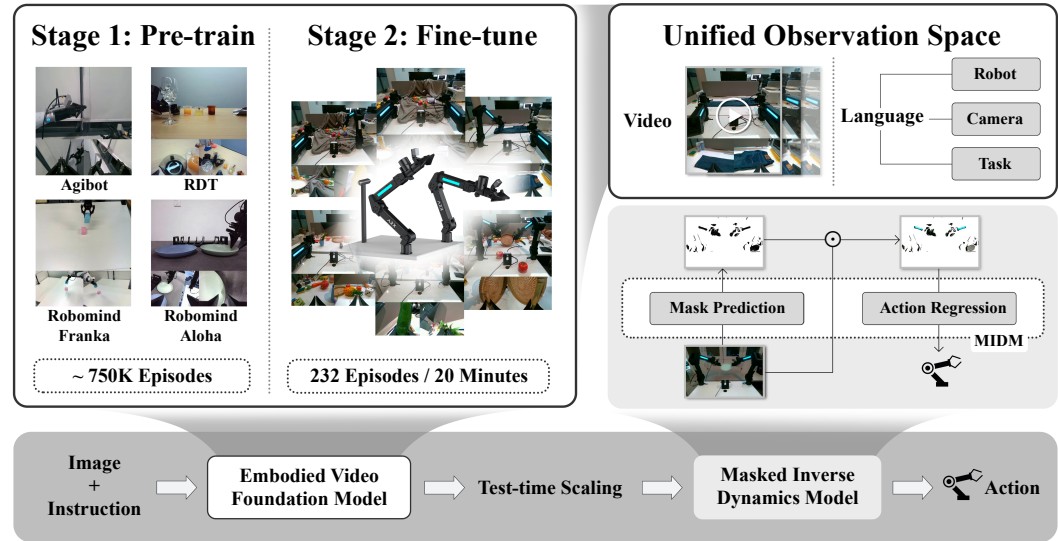

Figure 1: The overall pipeline of Vidar, where various video sources are leveraged for transferring to a new platform with limited demonstrations. A unified observation space handles heterogeneous, multi-view robotic videos and language instructions, enabling the pre-training of an embodied foundation video model on about 750,000 multi-view bimanual robotic episodes. After fine-tuning it with only 20 minutes of human demonstrations on an unseen robot platform, we adopt test-time scaling to select the best video during inference. Meanwhile, the masked inverse dynamics model (MIDM) converts videos to actions, where masks are learned to attend to action-relevant regions for background-robust action regression.

Here, $G$ is a video generation model that produces temporally coherent, physically plausible rollouts conditioned on task and observations, while $I$ is an inverse dynamics model that maps short video windows into robot-specific controls. This two-stage design shifts most of the representation burden to $G$, which can be pretrained on abundant Internet and robotic video, and leaves only a lightweight $I$ to be trained with limited demonstrations on the target platform.

Concretely, $G$ is conditioned on proprioceptive traces and embodiment tokens, and trained on 750K multi-view bimanual clips from three robot platforms, aligned into a unified observation space. This ensures that generated rollouts remain feasible under different morphologies and camera setups, and that contact and motion continuity are preserved. At inference, test-time scaling (Jaech et al., 2024) with physics-aware reranking further improves temporal coherence and physical plausibility. The inverse dynamics component $I$ is instantiated as a **Masked Inverse Dynamics Model (MIDM)**, which learns to attend to action-relevant regions such as hands, tools, and contact patches without pixel-level labels. By filtering out background clutter and distractors, MIDM provides robust action decoding and enables effective transfer of video priors across domains.

This formulation directly addresses the identified challenges: large-scale video pretraining reduces the need for triply-labeled demonstrations; conditioning and unified observation mitigate embodiment shifts; contact- and flow-consistent objectives make rollouts actionable; and MIDM grounds them to the target robot with few demonstrations. Together, these design choices turn raw video into transferable interaction priors that enable efficient and precise adaptation to new robotic embodiments.

## 2.2 VIDEO GENERATION MODEL

We adopt rectified flow (Lipman et al., 2023; Liu et al., 2023) models, which generate high-quality videos by modeling pixel flow over time. Specifically, the model parameterizes a flow function $v : \mathcal{V} \times \mathbb{R} \times \mathcal{L} \to \mathcal{V}$, which models the velocity of pixels as they transition from noisy video frames to target frames under certain conditions and time, leading to the following ODE over a video $x_t$

(with $x_0$ sampled from a Gaussian and $x_1$ as the output video):

$$\frac{\mathrm{d}x_t}{\mathrm{d}t} = v(x_t, t, c), \quad t \in [0, 1]. \tag{1}$$

To learn the non-trivial mapping between the Gaussian distribution and the video distribution, we train the "velocity" $v$ to approximate the constant flow from $x_0$ to $x_1$ during training. The flow matching loss is:

$$L_G = \mathbb{E}_{c,t,x_0,x_1} \left[ \|(x_1 - x_0) - v(tx_1 + (1-t)x_0, t, c)\|^2 \right]. \tag{2}$$

**Unified Observation Space.** To mitigate viewpoint and morphology gaps between heterogeneous embodiments, we design a unified observation space for multi-view embodied data. The space does not include actions: the video diffusion model only learns world evolution, allowing it to generalize efficiently across robots with different morphologies. Denoting the maximum number of cameras as $V$, we can define a unified observation space $\mathcal{U} \subseteq \mathcal{L} \times \mathcal{O}$ (also see Figure 1):

$$\mathcal{U} = \{\langle \mathbf{o}, \mathbf{l}\rangle | \mathbf{o} = \text{aggregate}(\mathbf{I}^{(1)}, \mathbf{I}^{(2)}, \cdots, \mathbf{I}^{(V)}), \mathbf{l} = \text{concatenate}(l_r, l_c, l_t)\}, \tag{3}$$

where $\text{aggregate}(\mathbf{I}^{(1)}, \mathbf{I}^{(2)}, \cdots, \mathbf{I}^{(V)})$ is the aggregation of image views (some views may be missing), and $l_r$, $l_c$, and $l_t$ are instructions related to the robotic platform, camera, and task, respectively. Specifically, each observation $\mathbf{o}$ is constructed as $\mathbf{o} = \bigoplus_{k=1}^{V} \phi_{r_k}(\mathbf{I}^{(k)})$, where $\mathbf{I}^{(k)}$ is the RGB image from camera $k$, $\phi_{r_k}$ is a spatial resizing function. This operation produces a consistent tensor shape across platforms and preserves both semantic and kinematic context. Instead of relying solely on task information and a single view, we condition the video generation model on the robot, camera, task information, and multiple views, thereby providing rich context for video prediction with unified representations across heterogeneous embodiments.

**Embodied Pre-training and Fine-tuning on Unified Observation Space.** We pre-train our video generation model on a large-scale corpus of approximately 750K open-sourced episodes, all projected into the unified observation space (Equation (3)). This dataset includes three camera views, a configuration commonly adopted in bimanual setups (Fu et al., 2024; AgiBot-World-Contributors et al., 2025) to provide abundant context information for precise control. At fine-tuning time, we apply supervised fine-tuning (SFT) on all model parameters using a small number of human demonstration episodes collected from the target platform. To ensure precise adaptation without overfitting, we augment the dataset by clipping variable-length videos from random starting points. In this way, the limited domain-specific data is fully used, and the model learns to predict videos at various states.

**Test-time Scaling.** Test-Time Scaling (TTS) refers to using additional test-time compute to improve performance without retraining the model, which is widely explored in large language models (Muennighoff et al., 2025). Diffusion-based video generation is inherently stochastic, resulting in significant variance in the quality, physical plausibility, and task relevance of sampled rollouts. Naïvely sampling a single trajectory may result in incoherent or suboptimal generations, especially in cluttered or ambiguous scenes. To address this, we propose a test-time scaling strategy: given an observation prefix $\mathbf{o}_1$, we generate $K$ candidate video trajectories $\{\tilde{\mathbf{v}}_{1:T}^{(i)}\}_{i=1}^{K}$ using different random seeds. We then rank these trajectories using a pretrained evaluator (e.g., CLIP or a vision-language model) $q_\eta$ and select the highest-scoring one, i.e., $\arg\max_i q_\eta(\tilde{\mathbf{v}}_{1:T}^{(i)})$. This approach aligns the predicted video distribution with task-relevant, actionable videos through "rejection sampling", reducing sampling variance and consistently improving the quality of generated demonstrations.

## 2.3 MASKED INVERSE DYNAMICS MODEL

Inverse dynamics models often suffer from poor generalization due to the presence of background noise, texture biases, and visual distractions in high-dimensional observations (Tan et al., 2025). Explicitly localizing action-relevant regions is challenging without dense annotations, and existing segmentation methods (Yuan et al., 2025) often fail to capture both arms in the bimanual settings, let alone produce temporally consistent segmentations (see Figure 6 in Appendix C). While prior work has explored information-theoretic formulations of state-space models (Nguyen et al., 2021), scaling to high dimensions remains challenging.

To solve the problem, we introduce a masked inverse dynamics model (MIDM) that learns to focus on task-relevant regions via implicit mask prediction. The model consists of two components: (1) a mask prediction network $U$ that outputs a spatial mask $m \in [0,1]^{H \times W}$ from an input frame $x$, and (2) an action regression network $R$ that predicts the action from the masked frame. Formally:

$$m = U(x), \quad \hat{a} = R(\text{Round}(m) \odot x),$$

where $\odot$ denotes element-wise multiplication, and Round(.) means rounding to the nearest integer. The model is trained by minimizing the following loss:

$$L_I = \mathbb{E}_{x,a} \left[ l(\hat{a} - a) + \lambda \|m\|_1 \right],$$

where $l(\cdot)$ is the Huber loss. The second $\ell_1$-norm regularization term promotes spatial sparsity, encouraging the model to focus on minimal, task-critical regions without any segmentation supervision. We train it using straight-through estimators. Notably, this framework is not restricted to predicting embodiment-specific actions; embodiment-agnostic actions can also be predicted.

By utilizing reliable action signals as supervision, rather than noisy annotations, this lightweight module demonstrates robust generalization to unseen environments and backgrounds with limited demonstrations, as evidenced by our experimental results (Section 3).

## 3 EXPERIMENTS

We now present experimental studies, with the goal to verify the following hypotheses:

**H1:** Vidar achieves superior success rates with only 20 minutes of target domain demonstrations;

**H2:** Vidar generalizes effectively to unseen tasks and backgrounds;

**H3:** Pre-training with a unified observation space benefits embodied video generation;

**H4:** Masked inverse dynamics models exhibit greater generalization ability than the baseline.

### 3.1 EXPERIMENTAL SETUP

#### 3.1.1 DATASETS

Our pre-training data integrates episodes sourced from Agibot-World (AgiBot-World-Contributors et al., 2025), RoboMind (Wu et al., 2024b), and RDT (Liu et al., 2024a). The resulting dataset for real-world experiments comprises 746,533 episodes. For the simulation experiments, we additionally add episodes from Egodex (Hoque et al., 2025).

For adaptation to *unseen* platforms, we collect target robot data in both the simulation and real-world domains. For the simulation domain, we employ two configurations: a low data setting and a standard one. In the low data setting, we collect 20 episodes per task using the Aloha (agilex) robot with an adjusted camera to fully capture the robot arms, under clean scenarios of the RoboTwin platform (Chen et al., 2025). In the standard data setting, we follow the leaderboard of RoboTwin by collecting 50 episodes per task with original camera viewpoints, where partial arm occlusion occurs more frequently. For the real-world domain, we collect 20 minutes of human demonstration videos, covering 81 tasks across 232 episodes.

All datasets collected consist of descriptions of the robot type and camera placements, as well as task instructions, as is required by the unified observation space. The collected embodiment-specific dataset is used both for fine-tuning the video diffusion model and training the masked inverse dynamics model. Notably, all these target domains are unseen during pre-training. Further details of our dataset can be found in Appendix A.

#### 3.1.2 TRAINING AND INFERENCE

We evaluate our method using two representative video generation models (with off-the-shelf checkpoints pre-trained on Internet-scale videos): the open-source Wan2.2 (Wang et al., 2025) for simulation experiments, and Vidu 2.0 (Bao et al., 2024) for the more diverse and challenging real-world tests. We use their pre-trained checkpoints and continue pre-training the models with a batch size of 128. The first 10,000 steps involve pre-training on the diverse robotics dataset, followed by 12,000

fine-tuning steps for Wan2.2 and 13,000 fine-tuning steps for Vidu 2.0. To reduce inference costs, we uniformly downsample the training videos to 8 frames per second (fps). For the masked inverse dynamics model, we adopt the U-Net (Ronneberger et al., 2015) structure as the mask prediction network and the ResNet (Xu et al., 2024) structure as the action regression network, with $\lambda = 3 \times 10^{-3}$ (effects of $\lambda$ are shown in Appendix C). We train it exclusively on the fine-tuning dataset. We use AdamW (Loshchilov & Hutter, 2019) for all our training. Here are some detailed settings of real-world experiments.

We use open-loop control for Vidar; the videos are generated in a single batch, without subsequent generation after the initial run. Using 8 NVIDIA Ampere-series 80GB GPUs, generating one video with 60 frames (7.5 seconds duration at 8fps) costs about 25 seconds. The time cost can be reduced using distillation or quantization, which are beyond the scope of this paper. For test-time scaling, we choose $K = 3$, and three videos with different random seeds are generated in parallel, evaluated by GPT-4o (Hurst et al., 2024). Additionally, we disable test-time scaling for simulation experiments for better reproducibility. More details of training and inference can be found in Appendix B.

### 3.1.3 BASELINES

For simulation experiments, we choose Pi0.5 (Intelligence et al., 2025) as our baseline. For real-world experiments, we perform preliminary experiments over multiple baselines and find that adaptation with only 20 minutes of videos and about 3 demonstrations per task is too challenging for vision-language-action models. Thus, we choose two baselines that also incorporate video-level prior knowledge: UniPi and VPP. To ensure fair comparison, we reproduce these methods over the advanced Vidu 2.0 model. We also compare Vidar (built on Wan 2.2) with Pi0.5 using a larger real-world dataset for completeness; the results are provided in Appendix D.

**Pi0.5.** We use the official checkpoint of Pi0.5, which has already been pre-trained on more than 10k hours of robot data, and follow the official framework to finetune the base model on multi-task demonstration data.

**UniPi.** We follow the official framework by fine-tuning the Vidu 2.0 model directly on our demonstration data and training an inverse dynamics model using a ResNet-based architecture.

**VPP.** We use the same checkpoint for the video generation model as Vidar, because VPP mainly focuses on how to use a given video generation model. Additionally, we train a diffusion model to generate short action sequences based on the latent features during one-step video generation and CLIP (Radford et al., 2021) embeddings of the task instructions. Notably, they use closed-loop control, which means new action sequences are generated and executed after previous executions.

### 3.2 EXPERIMENTAL RESULTS

**H1: Success Rates.** For the simulation experiments, we evaluate our method on the RoboTwin 2.0 (Chen et al., 2025) benchmark using the more challenging multi-task setting (in contrast, the official leaderboard trains a separate policy for each task), and the results for two data settings are summarized in Table 1. Compared to the strong baseline Pi0.5 (Intelligence et al., 2025), our method achieves a state-of-the-art average success rate across all scenarios. More detailed results can be found in Appendix C.

For real-world experiments, we test two baseline methods and our method under three scenarios: seen task and background (six tasks), unseen task (five tasks), and unseen background (six tasks). Detailed explanations are as follows:

- **Seen Tasks & Backgrounds**: three pick-and-place tasks (e.g., grasp the tomato using the left arm), two daily-life tasks (e.g., flip a dice using the right arm), and one bimanual task (lift the basket). The background we use is a cluttered office desk setup, featuring several computers situated behind the workspace.

- **Unseen Tasks**: three daily-life tasks (e.g., stack the bowl on the steamer using the left arm) and two semantic tasks (e.g., grasp the shortest bread using the left arm).

Table 1: Average success rates across 50 tasks, 100 episodes for the RoboTwin benchmark. Vidar consistently surpasses the strong Pi0.5 baseline across all settings and scenarios. The Pi0 results are taken directly from the official leaderboard, where each task is trained and evaluated independently under standard data settings, making them easier and not directly comparable.

| Data Regime | Low | | Standard | |
|---|---|---|---|---|
| Scenario | Clean | Randomized | Clean | Randomized |
| Pi0* | - | - | 46.42% | 16.34% |
| Pi0.5 | 25.0% | 9.2% | 44.8% | 14.2% |
| **Vidar (Ours)** | **60.0%** | **15.7%** | **65.8%** | **17.5%** |

- **Unseen Backgrounds**: two pick-and-place tasks (e.g., grasp a tomato using the left arm) and four daily-life tasks (e.g., flip a dice using the left arm). For unseen backgrounds, we include a studio setup with a typical green screen and a daily workspace setting with two cupboards containing robot supplies, which exhibit reflective surfaces.

The success rates are shown in Table 2. We find that our method achieves superior success rates over all three scenarios, demonstrating the effectiveness of our method. UniPi does not utilize heterogeneous robotic data, which restricts its generalization under limited demonstrations; VPP uses predicted features from a single denoising forward pass for action prediction, which leads to noise and instability—particularly in unseen environments. More details can be found in Appendix C.

Table 2: Success rates of different methods and configurations over robot manipulation tasks. Vidar achieves high success rates across all three scenarios, with great generalization ability to unseen tasks and backgrounds.

| Method | Seen Tasks & Backgrounds | Unseen Tasks | Unseen Backgrounds |
|---|---|---|---|
| VPP | 4.5% | 13.3% | 0.0% |
| UniPi | 36.4% | 6.7% | 22.2% |
| **Vidar (Ours)** | **68.2%** | **66.7%** | **55.6%** |

To further demonstrate the generality of our approach, we also perform additional real-world experiments using the open-sourced Wan2.2 and HunyuanVideo models (Kong et al., 2024). Notably, Vidar surpasses Pi0.5, achieving a 35% higher average success rate on the 7 seen tasks and a 54% higher success rate on the 7 unseen tasks. Related results are provided in Appendix D.

**H2: Generalization Ability.** For the simulation benchmark, we show that our model generalizes effectively to randomized scenarios, despite being trained only on clean scenarios (see Appendix C). For real-world experiments, quantitative results of our superior generation ability are shown in unseen scenarios of Table 2. We also provide some demonstrations in Figure 2, where Vidar demonstrates robust generalization to unseen tasks and backgrounds with strong semantic understanding. In Appendix E, we present more visualizations, including failure cases.

**H3: Effectiveness of Pre-training.** We evaluate the video generation quality of the original Vidu 2.0 model and our pre-trained version over the unseen target domain using VBench (Huang et al., 2024). As is shown in Table 3, we find that pre-training using large-scale robotic videos under our unified observation space enhances both the consistency and quality of generated frames, which are important for robot control tasks.

**H4: Effectiveness of MIDM.** Based on empirical observations of the Aloha robot's error tolerances, we define a successful prediction as having a maximum infinity norm error of less than 0.06 for joint positions and less than 0.6 for gripper positions, for both arms. Using this criterion, we evaluate the success rates of our masked inverse dynamics model (MIDM) and a ResNet baseline on both training and testing sets. The results, presented in Table 4, show that our MIDM demonstrates superior generalization compared to the baseline. Additionally, examples of the learned masks are provided in

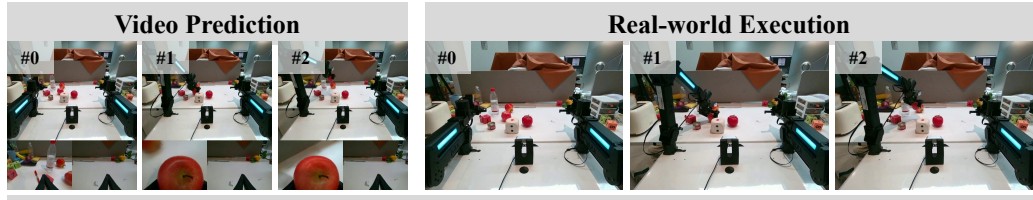

**Grasp the Red Apple:** There are multiple red objects on the table. The robot is instructed to grasp the red apple using its left arm. The robot needs to: **(0) identify the red apple among all red objects**, (1) secure its hold on the apple, (2) lift the apple as required.

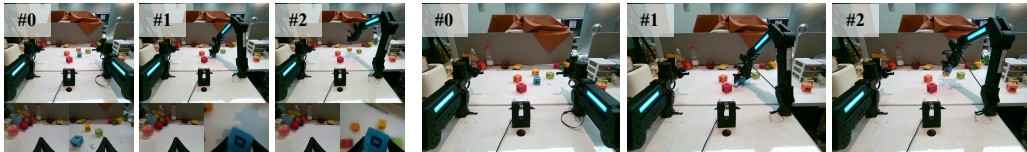

**Grasp the Blue Cube:** There are multiple cubes of different colors on the table. The robot is instructed to grasp the blue cube using its right arm. The robot needs to: **(0) identify the blue cube among all cubes**, (1) secure its hold on the cube, (2) lift the cube as required.

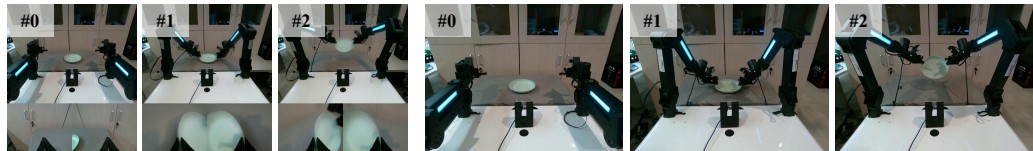

**Lift the dish:** There is **a green dish** on the **unseen table** in front of the **unseen cupboard**. The robot is instructed to lift the dish using both arms. The robot needs to: (0) align the grippers with the edge of the dish, (1) secure its hold on the dish, **(2) lift upwards simultaneously**.

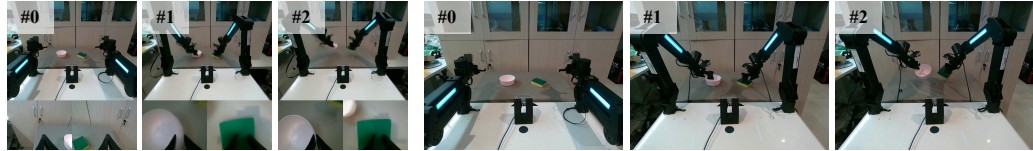

**Clean the bowl:** There is **a pink bowl and a sponge** on the **unseen table** in front of the **unseen cupboard**. The robot is instructed to clean the bowl using both arms. The robot needs to: (0) align the grippers with two objects, (1) secure its hold, **(2) lift upwards simultaneously**.

Figure 2: Videos of predictions (left) and corresponding executions (right) of Vidar for challenging tasks. It can handle unseen tasks and unseen backgrounds with strong semantic understanding.

Table 3: VBench video quality measurements for different video model configurations in the unseen target domain. Embodied pre-training over the unified observation space benefits video generation.

| Configuration | Subject Consistency | Background Consistency | Imaging Quality |
|---|---|---|---|
| Vidu 2.0 | 0.565 | 0.800 | 0.345 |
| **+ Embodied Pre-training** | **0.855** | **0.909** | **0.667** |

Figure 3. Without any additional supervision, MIDM effectively captures action-relevant features and generalizes well to unseen backgrounds.

## 3.3 ABLATION STUDY

We conduct an ablation study of our method by evaluating success rates on the same tasks presented in Table 2. The results are summarized in Table 5, where "w/o MIDM" means using the ResNet baseline. We find that both masked inverse dynamics models and test-time scaling are beneficial to the success rates.

Table 4: Training and testing success rates and testing $l_1$ errors of different inverse dynamics models. Both the baseline and MIDM achieve high performances during training, but MIDM generalizes better during test time.

| Inverse Dynamics Model | Training Accuracy | Testing Accuracy | Testing $l_1$ Error |
|---|---|---|---|
| ResNet | **99.9%** | 24.3% | 0.0430 |
| **MIDM (Ours)** | **99.9%** | **49.0%** | **0.0308** |

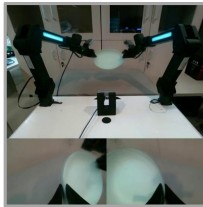 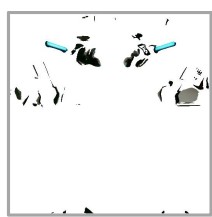 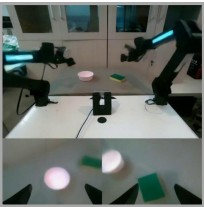 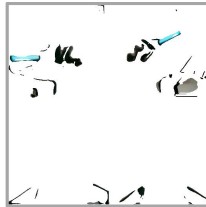

Figure 3: Input images and corresponding masked images learned by the masked inverse dynamic model (MIDM). The two cases are from an unseen background with complex reflective surfaces, while the predicted mask images still focus on the essential parts of robotic arms.

## 4 RELATED WORK

**Vision-Language-Action Models (VLAs).** Vision-Language-Action (VLA) models integrate perception, language understanding, and action generation to enable general-purpose embodied intelligence. However, existing VLA systems rely heavily on large-scale, task-specific datasets—often requiring hundreds of thousands of trajectories—to achieve robust performance. For instance, RT-1 (Brohan et al., 2023) uses 130K real-world episodes, while RT-2 (Zitkovich et al., 2023) scales to 1B image-text pairs. Recent works like OpenVLA (Kim et al., 2024), Octo (Ghosh et al., 2024), Pi0 (Black et al., 2024), and RDT-1B (Liu et al., 2024a) further expand to millions of demonstrations across diverse embodiments. Despite these efforts, such data-intensive pipelines present a scalability bottleneck and limit generalization to novel tasks and domains, motivating more data-efficient and transferable alternatives. However, actions are typically coupled to their VLA models, which constrains efficient transfer to heterogeneous embodiments.

**Coupled Video Generation and Action Synthesis.** Recent efforts couple video generation with action synthesis to enhance physical consistency and interoperability. For instance, Video-Prediction-Policy (VPP) (Hu et al., 2024) learns implicit inverse dynamics conditioned on future representations predicted by a video diffusion model, while UVA (Li et al., 2025) unifies video-action latent spaces with lightweight diffusion heads. Other works, such as VidMan (Wen et al., 2024), integrate low-level action representations with video prediction to predict actions. These approaches bridge spatiotemporal dynamics between vision and action, offering novel solutions for complex manipulation tasks. However, these approaches require end-to-end joint training of video generation and action prediction models, which limits their flexibility and adaptability.

Table 5: Ablation study of Vidar, where "w/o MIDM" means using the ResNet baseline. In three scenarios, both masked inverse dynamics models and test-time scaling are beneficial to the success rates.

| Configuration | Seen Tasks & Backgrounds | Unseen Tasks | Unseen Backgrounds |
|---|---|---|---|
| Vidar w/o TTS | 45.5% | 33.3% | 44.4% |
| Vidar w/o MIDM | 59.1% | 26.7% | 22.2% |
| **Vidar (Ours)** | **68.2%** | **66.7%** | **55.6%** |

**Video Generation Models for Embodied AI.** Motivated by one implementation of world models (Ha & Schmidhuber, 2018), video generation models for embodied AI also predict future scene dynamics to assist robotic planning and policy learning. Current works such as UniPi (Du et al., 2023), RoboDreamer (Zhou et al., 2024), Gen2Act (Bharadhwaj et al., 2024), CLOVER (Bu et al., 2024), SuSIE (Black et al., 2023), and GR-1 (Wu et al., 2024a) mainly adopt text-conditioned video generation or frame prediction with a fixed single-camera view, demonstrating how a single-arm robot physically compliantly executes a series of actions. Another orthogonal line of research, such as Dreamitate (Liang et al., 2024), focuses on tool use video predictions, where tools serve as mediators to simplify manipulation tasks. Additionally, Genie 2 (Parker-Holder et al., 2024) generates interactive 3D environments via autoregressive latent diffusion, enabling scalable training for embodied agents. These models implicitly encode physical laws through video synthesis, reducing reliance on real-world robotics data. However, current works either remain limited to single-arm robots and rely on a single camera view, or rely on the availability and functionality of specific tools, which restricts their applicability in more complex real-world scenarios; meanwhile, the success of Instant3D (Li et al., 2024) provides a compelling demonstration that the diffusion models can effectively adapt to multi-view formats. Moreover, most existing methods do not utilize heterogeneous embodied videos for pre-training.

## 5 CONCLUSION

We presented Vidar, a generalizable framework for bimanual robotic manipulation that addresses the core challenges of data scarcity and embodiment heterogeneity. By combining large-scale, diffusion-based video pretraining over a unified observation space with a masked inverse dynamics model, Vidar enables accurate action prediction from multi-view visual observations and language instructions, requiring only minimal demonstrations in new environments. Our experiments demonstrate that Vidar consistently outperforms existing methods and exhibits strong generalization to unseen tasks and backgrounds, highlighting its capacity for semantic understanding and transfer.

## 6 ETHICS STATEMENT

Vidar has the potential to accelerate the deployment of capable bimanual robots in real-world environments. However, the rise of generalist robotic systems also introduces important considerations regarding privacy, safety, and accountability, especially as these systems are deployed in sensitive domains involving close human-robot interaction.

## 7 REPRODUCIBILITY STATEMENT

We submit our code, including the HunyuanVideo diffusion model and the masked inverse dynamics model, in the supplemental materials. Meanwhile, the Wan2.2 model that we use is open-sourced and widely available. Appendix A describes our dataset in detail, and Appendix B provides comprehensive information on training and inference. All datasets used for pre-training are publicly available.

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

Table 6: Detailed information about datasets. Dataset instructions correspond to the robot and camera components of the unified observation space (see Figure 1). The platforms in the RoboTwin and the Vidar datasets are unseen during pre-training.

| Dataset | Dataset Instruction |
|---------|---------------------|
| Agibot | The whole scene is in a realistic, industrial art style with three views: **a fixed high camera, a movable left arm camera, and a movable right arm camera**. The **genie-1 robot** is currently performing the following task: |
| RDT | The whole scene is in a realistic, industrial art style with three views: **a fixed front camera, a movable left arm camera, and a movable right arm camera**. The **aloha robot** is currently performing the following task: |
| RoboMind Franka | The whole scene is in a realistic, industrial art style with three views: **a fixed camera on the opposite side, a fixed left camera, and a fixed right camera**. The **franka robot** is currently performing the following task: |
| RoboMind Aloha | The whole scene is in a realistic, industrial art style with three views: **a fixed front camera, a movable left arm camera, and a movable right arm camera**. The **aloha robot** is currently performing the following task: |
| Egodex | The whole scene is in a realistic, industrial art style with one view: **a movable front camera**. The **person** is currently performing the following task: |
| RoboTwin (Low Data) | The whole scene is in a realistic, industrial art style with three views: **a fixed rear camera, a movable left arm camera, and a movable right arm camera**. The **aloha robot** is currently performing the following task: |
| RoboTwin (Standard Data) | The whole scene is in a realistic, industrial art style with three views: **a fixed front camera, a movable left arm camera, and a movable right arm camera**. The **aloha robot** is currently performing the following task: |
| Vidar | The whole scene is in a realistic, industrial art style with three views: **a fixed rear camera, a movable left arm camera, and a movable right arm camera**. The **aloha robot** is currently performing the following task: |

## A  DATASET DETAILS

The details of our datasets are presented in Table 6. For the RoboTwin dataset, we adjust the camera positions to capture both arms in full, rather than only the end-effectors, to improve training of the masked inverse dynamics model. For the Agibot-World dataset, episodes are segmented into shorter clips using their frame-level annotations. We also filter out episodes with fewer than three views or shorter than four seconds. For each dataset, we provide information about the associated robot and camera setups, which form part of the unified observation space. We also leverage GPT-4o to augment its task annotations for the Vidar dataset. During pre-training, sampling ratios for each dataset are set proportionally to their sizes.

It is worth noting that the RoboTwin dataset and our Vidar dataset differ from all pre-training datasets in these aspects: for example, beyond the use of different robot arms, the central camera is positioned far behind and high above the scene in the Vidar dataset—significantly different from the views in the pre-training datasets.

> You are a skilled robot video ranker. Your task is to identify the index of the video with the highest quality based on the provided image clips and video caption. When evaluating the images, consider both their physical accuracy and how well they align with the video caption. Each image contains three views, and you must assess their consistency, ensuring there are no abrupt appearances or disappearances of objects or color blocks between frames. When determining the index, if there is a tie, output the smallest video index. For example: if video 1 has the highest quality, output 1; if video 2 has the highest quality, output 2; if all the videos have the same quality, output 1. We have {n_videos} videos, each containing {n_imgs_per_video} images, for you to evaluate. The caption for video_reference is '{caption}'. The images are arranged in the following sequence:
> {img_seq}
> Please assess the quality of the videos and provide the index of the one with the highest quality, without any explanations.

Figure 4: Prompt for GPT-4o evaluation. Variables in the curly braces should be replaced by corresponding values. Specifically, one line of "img_seq" describes one video, and is formatted as "-**video_1**: image_1, image_2, ..., image_{n_imgs_per_video}".

Table 7: Hyperparameters of MIDM.

| Hyperparameter | Value |
|---|---|
| Number of Parameters | 92 Million |
| U-Net Down-sampling/Up-sampling Layers | 5 |
| ResNet Structure | ResNet-50 |
| Action Prediction Loss | Huber Loss |
| Learning Rate | $5 \times 10^{-4}$ |
| Warm-up | 6000 Steps |
| AdamW $\beta$ | $(0.9, 0.999)$ |
| AdamW $\epsilon$ | $10^{-8}$ |
| AdamW Weight Decay | $10^{-2}$ |

## B  TRAINING AND INFERENCE DETAILS

Using 64 NVIDIA Ampere-series 80GB GPUs, we train Vidu 2.0 for 23,000 iterations (10,000 for pre-training and the remainder for fine-tuning), which takes about 64 hours. Note that for all fine-tuning procedures, we employ full-parameter fine-tuning. For MIDM, we use 8 NVIDIA Hopper-series 80GB GPUs for 60,000 training iterations, taking about 5 hours. Additional MIDM hyperparameters are provided in Table 7. We trained Pi0.5 model as a baseline for 55,000 iterations for real-world tasks, with action horizon chosen to be 16. Additional Pi0.5 hyperparameters are provided in Table 8

During inference, the video diffusion models are deployed in the cloud, while only the lightweight MIDM is executed locally. For test-time scaling, we uniformly sample $5-7$ frames from the generated videos and use GPT-4o to select the best result. The prompt we use is shown in Figure 4, focusing on physical plausibility and alignment with the textual instruction. Each pairwise comparison costs about \$0.003, and it accounts for about 25% of the total latency when $K = 3$.

## C  EXPERIMENTAL RESULTS

For the simulation experiments, we set $K = 1$ (i.e., no test-time scaling) for better reproducibility. During testing, we limit the maximum steps to 180, which means there are three model inferences with 60 steps each. We use Pi0.5 as our baseline and trained both Vidar and Pi0.5 under two data regimes. Specifically, under the standard-data setting, models are trained under the clean scenario with 50 episodes for each task; under the low-data setting, models are trained under clean scenario with 20 episodes with adjusted camera views to test their adaptivity to different views. Notably,

Table 8: Hyperparameters of Pi0.5.

| Hyperparameter | Value |
|---|---|
| Number of Parameters | 2 Billion |
| Learning Rate | $2.5 \times 10^{-5}$ |
| Batch Size | 32 |
| Warm-up | 1000 Steps |
| Optimizer | AdamW |
| AdamW $\beta$ | $(0.9, 0.95)$ |
| AdamW $\epsilon$ | $10^{-8}$ |
| AdamW Weight Decay | $10^{-10}$ |

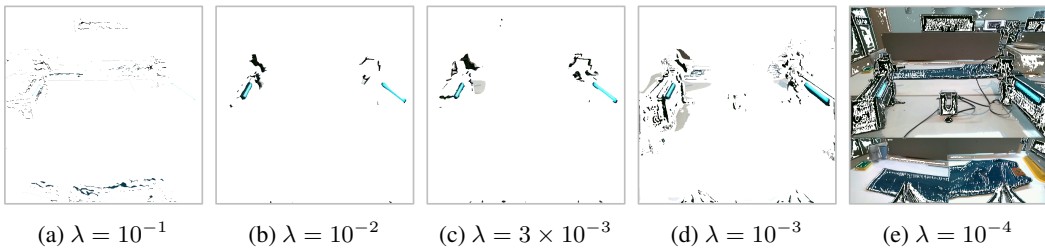

(a) $\lambda = 10^{-1}$  (b) $\lambda = 10^{-2}$  (c) $\lambda = 3 \times 10^{-3}$  (d) $\lambda = 10^{-3}$  (e) $\lambda = 10^{-4}$

Figure 5: Masked images learned by the masked inverse dynamic model (MIDM) with different values of $\lambda$.

we adopt a multi-task setting instead of training the model separately for each task, which is more challenging. The testing success rates averaged over 100 episodes are shown in Table 9 and Table 10.

For the real-world experiments, a more detailed version of Table 2 is provided in Table 11. We also investigate the impact of the hyperparameter $\lambda$ in MIDM, with results summarized in Table 12 and illustrated in Figure 5. Notably, $\lambda = 3 \times 10^{-3}$ yields the best performance.

We also test the performance of an existing segmentation model, RoboEngine (Yuan et al., 2025). As is shown in Figure 6, it often identifies only one arm per frame, fails to recognize grippers in wrist camera views, or lacks temporal consistency across frames.

# D  ADDITIONAL REAL-WORLD EXPERIMENTS

We also run additional real-world experiments using the Wan 2.2 model and the HunyuanVideo model as our backbone. The training hyperparameters are detailed in Table 13. Utilizing 64 NVIDIA Hopper-series 80GB GPUs, the training of the HunyuanVideo Model required approximately 54 hours.

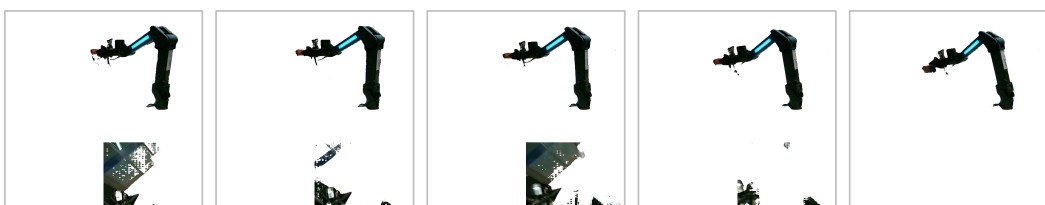

Figure 6: Segmentation results. We split the concatenated video frames into three views and apply RoboEngine (Yuan et al., 2025) to each view.

Table 9: Success rates of Vidar and Pi0.5 on the RoboTwin 2.0 benchmark under clean scenario.

| Data Regime | Low | | Standard | |
| Task | Pi0.5 | Vidar | Pi0.5 | Vidar |
|---|---|---|---|---|
| Adjust Bottle | 95.0% | 100.0% | 98.0% | 63.0% |
| Beat Block Hammer | 40.0% | 85.0% | 54.0% | 93.0% |
| Blocks Ranking RGB | 11.0% | 55.0% | 25.0% | 52.0% |
| Blocks Ranking Size | 1.0% | 35.0% | 10.0% | 21.0% |
| Click Alarmclock | 59.0% | 100.0% | 93.0% | 95.0% |
| Click Bell | 63.0% | 95.0% | 92.0% | 100.0% |
| Dump Bin Bigbin | 35.0% | 50.0% | 49.0% | 72.0% |
| Grab Roller | 81.0% | 100.0% | 96.0% | 96.0% |
| Handover Block | 2.0% | 5.0% | 4.0% | 2.0% |
| Handover Mic | 17.0% | 0.0% | 31.0% | 24.0% |
| Hanging Mug | 0.0% | 0.0% | 8.0% | 1.0% |
| Lift Pot | 14.0% | 90.0% | 3.0% | 93.0% |
| Move Can Pot | 18.0% | 60.0% | 23.0% | 48.0% |
| Move Pillbottle Pad | 11.0% | 70.0% | 31.0% | 72.0% |
| Move Playingcard Away | 58.0% | 100.0% | 74.0% | 97.0% |
| Move Stapler Pad | 1.0% | 35.0% | 19.0% | 28.0% |
| Open Laptop | 42.0% | 50.0% | 46.0% | 73.0% |
| Open Microwave | 11.0% | 20.0% | 21.0% | 43.0% |
| Pick Diverse Bottles | 20.0% | 55.0% | 24.0% | 67.0% |
| Pick Dual Bottles | 23.0% | 85.0% | 54.0% | 87.0% |
| Place A2B Left | 5.0% | 45.0% | 53.0% | 86.0% |
| Place A2B Right | 3.0% | 55.0% | 43.0% | 91.0% |
| Place Bread Basket | 33.0% | 75.0% | 46.0% | 82.0% |
| Place Bread Skillet | 1.0% | 85.0% | 41.0% | 79.0% |
| Place Burger Fries | 28.0% | 80.0% | 78.0% | 93.0% |
| Place Can Basket | 19.0% | 50.0% | 25.0% | 38.0% |
| Place Cans Plasticbox | 4.0% | 0.0% | 17.0% | 69.0% |
| Place Container Plate | 62.0% | 100.0% | 80.0% | 98.0% |
| Place Dual Shoes | 1.0% | 0.0% | 4.0% | 9.0% |
| Place Empty Cup | 20.0% | 100.0% | 95.0% | 92.0% |
| Place Fan | 8.0% | 45.0% | 21.0% | 55.0% |
| Place Mouse Pad | 3.0% | 60.0% | 19.0% | 74.0% |
| Place Object Basket | 31.0% | 35.0% | 66.0% | 55.0% |
| Place Object Scale | 14.0% | 85.0% | 40.0% | 75.0% |
| Place Object Stand | 36.0% | 95.0% | 64.0% | 90.0% |
| Place Phone Stand | 19.0% | 75.0% | 30.0% | 82.0% |
| Place Shoe | 13.0% | 80.0% | 61.0% | 89.0% |
| Press Stapler | 58.0% | 90.0% | 80.0% | 98.0% |
| Put Bottles Dustbin | 0.0% | 0.0% | 11.0% | 3.0% |
| Put Object Cabinet | 0.0% | 0.0% | 33.0% | 22.0% |
| Rotate QRcode | 27.0% | 65.0% | 51.0% | 65.0% |
| Scan Object | 4.0% | 45.0% | 17.0% | 47.0% |
| Shake Bottle | 86.0% | 100.0% | 93.0% | 99.0% |
| Shake Bottle Horizontally | 72.0% | 100.0% | 100.0% | 99.0% |
| Stack Blocks Three | 4.0% | 15.0% | 9.0% | 25.0% |
| Stack Blocks Two | 37.0% | 80.0% | 57.0% | 90.0% |
| Stack Bowls Three | 3.0% | 45.0% | 37.0% | 39.0% |
| Stack Bowls Two | 22.0% | 95.0% | 75.0% | 92.0% |
| Stamp Seal | 7.0% | 50.0% | 28.0% | 68.0% |
| Turn Switch | 27.0% | 60.0% | 10.0% | 60.0% |
| Average | 25.0% | 60.0% | 44.8% | 65.8% |

Table 10: Success rates of Vidar and Pi0.5 on the RoboTwin 2.0 benchmark under randomized scenario.

| Data Regime | Low | | Standard | |
| Task | Pi0.5 | Vidar | Pi0.5 | Vidar |
|---|---|---|---|---|
| Adjust Bottle | 16.0% | 65.0% | 35.0% | 39.0% |
| Beat Block Hammer | 5.0% | 10.0% | 5.0% | 10.0% |
| Blocks Ranking RGB | 0.0% | 0.0% | 0.0% | 4.0% |
| Blocks Ranking Size | 0.0% | 0.0% | 0.0% | 0.0% |
| Click Alarmclock | 30.0% | 35.0% | 67.0% | 74.0% |
| Click Bell | 16.0% | 25.0% | 42.0% | 68.0% |
| Dump Bin Bigbin | 16.0% | 10.0% | 25.0% | 10.0% |
| Grab Roller | 28.0% | 30.0% | 34.0% | 37.0% |
| Handover Block | 2.0% | 0.0% | 0.0% | 0.0% |
| Handover Mic | 0.0% | 0.0% | 4.0% | 8.0% |
| Hanging Mug | 0.0% | 0.0% | 2.0% | 0.0% |
| Lift Pot | 0.0% | 10.0% | 0.0% | 4.0% |
| Move Can Pot | 4.0% | 0.0% | 4.0% | 0.0% |
| Move Pillbottle Pad | 6.0% | 20.0% | 2.0% | 4.0% |
| Move Playingcard Away | 19.0% | 40.0% | 13.0% | 22.0% |
| Move Stapler Pad | 0.0% | 0.0% | 6.0% | 7.0% |
| Open Laptop | 26.0% | 30.0% | 2.0% | 24.0% |
| Open Microwave | 4.0% | 0.0% | 8.0% | 5.0% |
| Pick Diverse Bottles | 10.0% | 0.0% | 6.0% | 9.0% |
| Pick Dual Bottles | 17.0% | 15.0% | 6.0% | 30.0% |
| Place A2B Left | 1.0% | 10.0% | 7.0% | 7.0% |
| Place A2B Right | 0.0% | 15.0% | 7.0% | 17.0% |
| Place Bread Basket | 10.0% | 15.0% | 15.0% | 7.0% |
| Place Bread Skillet | 0.0% | 10.0% | 12.0% | 8.0% |
| Place Burger Fries | 14.0% | 5.0% | 47.0% | 11.0% |
| Place Can Basket | 4.0% | 0.0% | 2.0% | 4.0% |
| Place Cans Plasticbox | 0.0% | 0.0% | 11.0% | 13.0% |
| Place Container Plate | 38.0% | 55.0% | 31.0% | 23.0% |
| Place Dual Shoes | 0.0% | 0.0% | 0.0% | 3.0% |
| Place Empty Cup | 4.0% | 20.0% | 32.0% | 33.0% |
| Place Fan | 0.0% | 0.0% | 2.0% | 10.0% |
| Place Mouse Pad | 0.0% | 10.0% | 1.0% | 14.0% |
| Place Object Basket | 5.0% | 10.0% | 6.0% | 4.0% |
| Place Object Scale | 1.0% | 0.0% | 9.0% | 13.0% |
| Place Object Stand | 22.0% | 35.0% | 10.0% | 10.0% |
| Place Phone Stand | 7.0% | 25.0% | 3.0% | 18.0% |
| Place Shoe | 6.0% | 40.0% | 13.0% | 24.0% |
| Press Stapler | 13.0% | 40.0% | 58.0% | 48.0% |
| Put Bottles Dustbin | 0.0% | 0.0% | 3.0% | 0.0% |
| Put Object Cabinet | 0.0% | 0.0% | 1.0% | 3.0% |
| Rotate QRcode | 4.0% | 10.0% | 0.0% | 1.0% |
| Scan Object | 0.0% | 5.0% | 1.0% | 6.0% |
| Shake Bottle | 56.0% | 65.0% | 67.0% | 75.0% |
| Shake Bottle Horizontally | 46.0% | 60.0% | 59.0% | 73.0% |
| Stack Blocks Three | 0.0% | 0.0% | 0.0% | 3.0% |
| Stack Blocks Two | 8.0% | 5.0% | 9.0% | 16.0% |
| Stack Bowls Three | 0.0% | 15.0% | 10.0% | 4.0% |
| Stack Bowls Two | 6.0% | 35.0% | 31.0% | 30.0% |
| Stamp Seal | 1.0% | 0.0% | 4.0% | 7.0% |
| Turn Switch | 17.0% | 10.0% | 0.0% | 33.0% |
| Average | 9.2% | 15.7% | 14.2% | 17.5% |

Table 11: Detailed success rates of various methods and configurations on robot manipulation tasks. "L", "R", and "B" indicate the use of the left arm, right arm, or both arms, respectively. "w/o TTS" and "w/o MIDM" correspond to the ablation settings described in Table 5.

| Scenario/Task | Success Rate | | | | |
|---|---|---|---|---|---|
| Seen Tasks & Backgrounds | UniPi | VPP | **Vidar** | w/o TTS | w/o MIDM |
| Grasp the Tomato (L) | 60.0% | 0.0% | 60.0% | 60.0% | 80.0% |
| Grasp the Tomato (R) | 20.0% | 0.0% | 80.0% | 60.0% | 60.0% |
| Lift the Basket (B) | 66.7% | 0.0% | 66.7% | 33.3% | 33.3% |
| Flip the Dice to Point One on the Top (L) | 0.0% | 33.3% | 33.3% | 0.0% | 33.3% |
| Grab the Bottle (L) | 0.0% | 0.0% | 66.7% | 33.3% | 33.3% |
| Get the Toast from the Toaster (L) | 66.7% | 0.0% | 100.0% | 66.7% | 100.0% |
| Average | 36.4% | 4.5% | **68.2%** | 45.5% | 59.1% |
| Unseen Tasks | UniPi | VPP | **Vidar** | w/o TTS | w/o MIDM |
| Place the Bowl on the Steamer (L) | 0.0% | 0.0% | 66.7% | 66.7% | 33.3% |
| Grasp the Shortest Bread (L) | 0.0% | 0.0% | 66.7% | 33.3% | 0.0% |
| Grasp the Shortest Bread (R) | 0.0% | 0.0% | 66.7% | 33.3% | 33.3% |
| Wipe the Table with Rag (L) | 33.3% | 33.3% | 66.7% | 33.3% | 66.7% |
| Wipe the Table with Rag (R) | 0.0% | 33.3% | 66.7% | 0.0% | 0.0% |
| Average | 6.7% | 13.3% | **66.7%** | 33.3% | 26.7% |
| Unseen Backgrounds | UniPi | VPP | **Vidar** | w/o TTS | w/o MIDM |
| Grasp the Tomato (L) | 0.0% | 0.0% | 66.7% | 33.3% | 0.0% |
| Flip the Die to Point One on the Top (L) | 0.0% | 0.0% | 33.3% | 33.3% | 0.0% |
| Flip the Die to Point One on the Top (R) | 0.0% | 0.0% | 33.3% | 0.0% | 0.0% |
| Pick the Carrot on the Green Plate (L) | 33.3% | 0.0% | 33.3% | 66.7% | 0.0% |
| Place the Bowl on the Plate (L) | 33.3% | 0.0% | 66.7% | 66.7% | 33.3% |
| Place the Bowl on the Plate (R) | 66.7% | 0.0% | 100.0% | 66.7% | 100.0% |
| Average | 22.2% | 0.0% | **55.6%** | 44.4% | 22.2% |

Table 12: Analysis of hyperparameter $\lambda$ in the masked inverse dynamics model. The success rates are robust over a large range, and $\lambda = 3 \times 10^{-3}$ achieves the best performance.

| $\lambda$ | Training Accuracy | Testing Accuracy | Testing $l_1$ Error |
|---|---|---|---|
| $10^{-1}$ | 99.1% | 7.1% | 0.0670 |
| $10^{-2}$ | 99.8% | 39.9% | 0.0331 |
| $3 \times 10^{-3}$ | **99.9%** | **49.0%** | **0.0308** |
| $10^{-3}$ | 99.9% | 40.7% | 0.0338 |
| $10^{-4}$ | 99.8% | 24.4% | 0.0461 |

For the Wan 2.2 model, we evaluate it alongside Pi0.5 (also trained for 55,000 iterations). Since Pi0.5 is typically fine-tuned on a larger dataset, we expand the fine-tuning set to 2,307 episodes. We then evaluate both models on 14 tasks (7 seen and 7 unseen), with results reported in Table 14. As shown, Vidar consistently outperforms Pi0.5 on average.

We evaluate the HunyuanVideo model version on six tasks. The results are presented in Table 15 and Figure 7. We plan to open-source our implementations related to both the HunyuanVideo Model and our MIDM.

## E  ADDITIONAL VISUALIZATIONS

More demonstrations, including two failed cases, are shown in Figure 8.

Table 13: Hyperparameters of the Wan 2.2 model and the HunyuanVideo Model.

| Hyperparameter | Wan2.2 | HunyuanVideo |
|---|---|---|
| Number of Parameters | 5 Billion | 13 Billion |
| Learning Rate (Pre-train) | $2 \times 10^{-5}$ | $1 \times 10^{-4}$ |
| Learning Rate (Fine-tune) | $2 \times 10^{-5}$ | $5 \times 10^{-5}$ |
| Warm-up | 200 Steps | 500 steps |
| Optimizer | AdamW | AdamW |
| AdamW $\beta$ | $(0.9, 0.999)$ | $(0.9, 0.999)$ |
| AdamW $\epsilon$ | $10^{-8}$ | $10^{-8}$ |
| AdamW Weight Decay | 0.1 | 0 |
| Pre-training Steps | 10,000 | 10,000 |
| Fine-tuning Steps | 12,000 | 2,000 |

Table 14: Success rates of Vidar reproduced using the Wan2.2 Model. "L", "R", and "B" indicate the use of the left arm, right arm, or both arms, respectively.

| Seen Cases | Vidar | Pi0.5 |
|---|---|---|
| Flip the Die to Show One on Its Top (L) | 40.0% | 20.0% |
| Grasp the Fruit from the Basket (R) | 60.0% | 0.0% |
| Click the Mouse's Left Button (L) | 80.0% | 0.0% |
| Lift the Basket (B) | 80.0% | 20.0% |
| Pour Water into the Cup (L) | 100.0% | 25.0% |
| Wipe the Table with a Blue Rag (L) | 100.0% | 100.0% |
| Grab the Blue Cup (R) | 25.0% | 75.0% |
| Average | 69.3% | 34.3% |
| Unseen Cases | Vidar | Pi0.5 |
| Close the Laptop (L) | 80.0% | 0.0% |
| Close the Laptop (R) | 60.0% | 0.0% |
| Throw the Paper Ball (L) | 80.0% | 40.0% |
| Shake the Water Bottle (R) | 60.0% | 0.0% |
| Grasp and Place Two Breads (B) | 90.0% | 20.0% |
| Wipe the Desk with a Blue Towel (B) | 20.0% | 0.0% |
| Manipulate the Controller Handle (L) | 80.0% | 30.0% |
| Average | 67.1% | 12.9% |

Table 15: Success rates of Vidar reproduced using the HunyuanVideo model.

| Task | Success Rate |
|---|---|
| Grasp the Apple (Seen Task) | 75.0% |
| Grab the Bottle (Seen Task) | 100.0% |
| Grasp the Cup by its Handle (Seen Task) | 25.0% |
| Lift the Steamer (Unseen Task) | 50.0% |
| Grasp the Apple and Put it into the Steamer (Unseen Task) | 100.0% |
| Stack One Cube on Top of the Other Cube (Unseen Task) | 20.0% |
| Average | 58.3% |

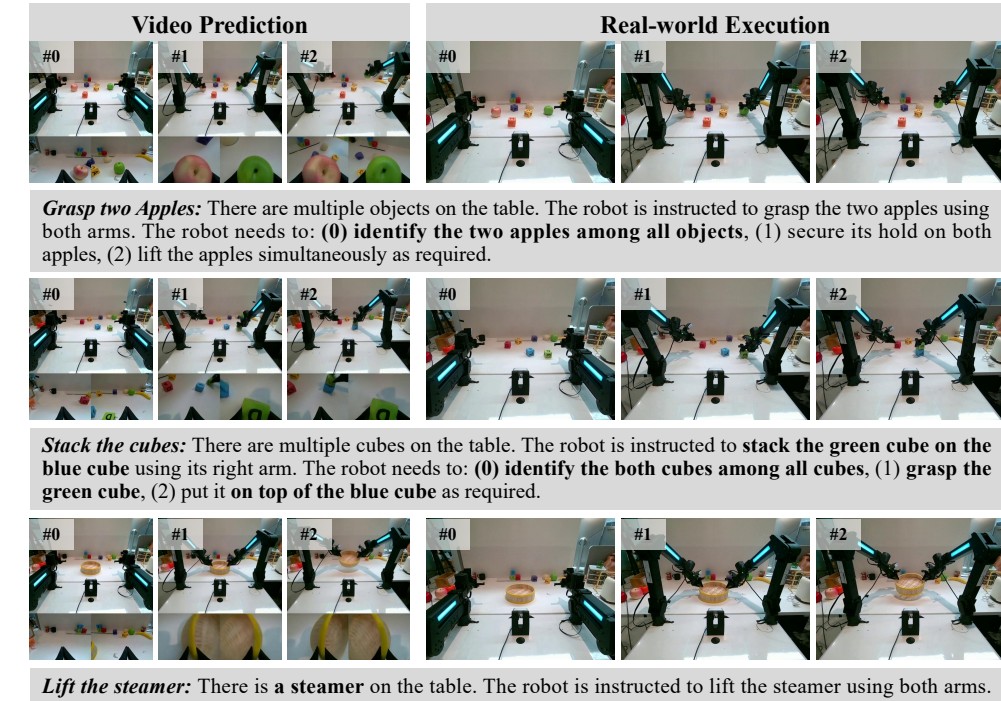

**Grasp two Apples:** There are multiple objects on the table. The robot is instructed to grasp the two apples using both arms. The robot needs to: **(0) identify the two apples among all objects**, (1) secure its hold on both apples, (2) lift the apples simultaneously as required.

**Stack the cubes:** There are multiple cubes on the table. The robot is instructed to **stack the green cube on the blue cube** using its right arm. The robot needs to: **(0) identify the both cubes among all cubes**, (1) **grasp the green cube**, (2) put it **on top of the blue cube** as required.

**Lift the steamer:** There is **a steamer** on the table. The robot is instructed to lift the steamer using both arms. The robot needs to: (0) align the grippers with the edge of the steamer, (1) secure its hold on the steamer, **(2) lift upwards simultaneously**.

Figure 7: Videos of predictions (left) and corresponding executions (right) of Vidar reproduced using the HunyuanVideo model for challenging tasks.

## F    HARDWARE DETAILS

Hardware details are shown in Figure 9 and Table 16. One important assumption underlying this approach is that the intermediate video modality contains all the information for action prediction. However, this assumption fails to hold for many robotic systems due to the specific platform setting, including the Aloha platform. In the pre-training part of Figure 1, we can see that the arm joints frequently fall outside the camera's field of view, even with three different views available. In our target domain, we adjust the center camera to a new position where two arms can be fully captured, shown in the fine-tuning part of Figure 1. In this way, our robotic platform differs from any platform we encountered during pre-training, serving as an ideal testbed for showing adaptation capability with scarce data.

## G    LARGE LANGUAGE MODEL USAGE

We utilize large language models to refine the writing. For example, we write a draft and let the model check the grammar errors.

| Video Prediction | Real-world Execution |
|---|---|

*Flip the die:* There is **a die** on the **unseen table** in front of the **unseen cupboard**. The robot is instructed to flip the die to orient face one upward using its right arm. The robot needs to: **(0) identify the single-pip face**, (1) secure its hold on the die, (2) **flip the die** as required.

*Wipe the table with rag:* There is **a rag** on the table. The robot is instructed to wipe table with rag using its right arm. The robot needs to: **(0) identify the rag and predict its motion trajectory**, (1) secure its hold on the rag, (2) **move the rag leftward and rightward** to wipe table as required.

*Stack the bowl on steamer:* There is **a pink bowl and a steamer** on the **unseen table** in front of the **unseen cupboard**. The robot is instructed to stack the bowl on steamer using its left arm. The robot needs to: (0) align the grippers with the edge of the bowl, (1) secure its hold on the bowl, **(2) move the bowl onto the steamer**.

*Stack the bowl on steamer (failure):* There is **a pink bowl and a steamer** on the **unseen table** in front of the **unseen cupboard**. The robot is instructed to stack the bowl on steamer using its left arm. The robot needs to: (0) align the grippers with the edge of the bowl, (1) secure its hold on the bowl, **(2) move the bowl onto the steamer**(failed due to motion prediction misalignment).

*Lift the dish (failure):* There is **a green dish** on the **unseen table** in front of the **unseen cupboard**. The robot is instructed to lift the dish using both arms. The robot needs to: (0) align the grippers with the edge of the dish, (1) secure its hold on the dish(failed due to gripper mechanical misalignment), **(2) lift upwards simultaneously**.

Figure 8: Videos of predictions (left) and corresponding executions (right) of Vidar for more challenging tasks. Both successful and failed cases are presented.

Figure 9: Our robotic platform.

Table 16: Hardware Information.

| Parameter | Value |
|---|---|
| Degree of Freedom | $2 \times (6 + 1) = 14$ |
| Cameras | 3 RGB Cameras |
| Arm weight | $3.9\,\mathrm{kg}$ |
| Arm Valid Payload | $1.0\,\mathrm{kg}$ |
| Arm Reach | $0.6\,\mathrm{m}$ |
| Arm Repeatability | $1\,\mathrm{mm}$ |
| Gripper Range | 0 - 80 mm |
| Gripper Max Force | $10\,\mathrm{N}$ |

