# OpenReview forum: "Vidar: Embodied Video Diffusion Model for Generalist Manipulation"
_ICLR.cc/2026/Conference — Submitted to ICLR 2026_

### Official Review · Reviewer_8PBm · 2025-10-30

**Soundness:** 3
**Presentation:** 3
**Contribution:** 3
**Rating:** 6
**Confidence:** 3

**Summary:**

The paper aims to achieve generalist few-shot policy learner by leveraging embodiment agnostic priors such as video generative models. Specifically, the authors decompose the generalist robot policy as a composition of embodiment agnostic video predictive prior and embodiment specific adaptor. To improve the data efficiency and generalizability, a sparse mask-based regularization is proposed. Instead of naively conditioning the embodiment specific adaptor, the proposed method learns a parsimonious mask. As can be seen from the provided videos and figures, this enables the model to correctly filter out pixels that are irrelevant to predicting the grounded low-level action. Experimental results demonstrate this omission of information indeed helps downstream adaptation.

**Strengths:**

The qualitative visual results clearly show that the learned model can successfully point out the pixels that are relevant to predicting the low level policy. The methodology and claimed benefits are sound, and these are also well supported quantitatively through several experiments.

**Weaknesses:**

* This kind of model falls into a broader category of learning unified action/observation representation. Comparing the proposed mask-based approach with other embodiment-agnostic action representation such as flow-based ones [1] would further strengthen the paper's argument.

* Several works have already explored filtering out action-irrelevant information from visual inputs. For instance, [2] proposed more general information-theoretic approach that is not limited to pixel-aligned masking.

[1] "General Flow as Foundation Affordance for Scalable Robot Learning," Yuan et al., 2024

[2] "Temporal Predictive Coding For Model-Based Planning In Latent Space," Nguyen et al., 2021

**Questions:**

How sensitive is this method to different hyperparameters?

---

> ### Author Response · Authors · 2025-11-20
> **Response to Reviewer 8PBm**
>
> We thank the reviewer for the thoughtful comments and for highlighting the strengths of our work. Below, we address the weaknesses and questions in detail.
>
> ## **Weaknesses**
> ### **1. Comparison with Prior Embodiment-agnostic Action Representations**
> Thank you for raising this important point. We agree that unified action/observation representations—such as the flow-based formulation in [1]—form a valuable direction for scalable robot learning. Meanwhile, our mask-based mechanism can be built on top of any action representation—whether embodiment-agnostic or not—and is largely **orthogonal** to these approaches. For example, instead of using heuristic selection in [1], our mask mechanism can be integrated to guide querying the flow model. We believe the two directions are complementary, and combining them could further enhance both interpretability and performance.
>
> ### **2. Relation to Prior Work on Filtering Action-irrelevant Information**
> We appreciate the reviewer pointing out this related direction. The information-theoretic framework in [2] is a powerful and general approach, particularly when built upon RSSM-style latent dynamics. Compared to such methods, our approach offers three practical advantages:
> - **Scalability**: Estimating mutual information in high-dimensional spaces is inherently challenging, whereas our method remains effective directly in the high-dimensional image space.
> - **Lightweight Integration:** Our masking mechanism is simple to train, does not require modifying the video diffusion model or dynamics, and can be easily added to existing policy architectures.
> - **Interpretability and Reusability:** The learned pixel-aligned masks are directly visualizable, making it easy to understand what information the adaptor relies on. Moreover, these masks can be **reused**—for example, to improve video diffusion prediction by guiding attention toward task-relevant regions.
>
> Taken together, the simplicity and interpretability make our mechanism a complementary tool to more general information-theoretic methods.
>
> ## **Questions**
> ### **1. Hyperparameter Sensitivity**
> We provide hyperparameter sensitivity analyses in the main text and appendix:
>
> - **Mask Sparsity Weight $\lambda$:**
>   The effects are detailed in Figure 5 and Table 8. We observe that performance degrades only gradually when $\lambda$ is varied across a reasonably broad range.
>
> - **Test-time Scaling:**
>   The ablations in Table 4 indicate that the model's performance remains relatively stable across different test-time scaling configurations.
>
> Due to the substantial computational cost—particularly for training the video diffusion prior—we could not perform exhaustive sweeps. However, within the practical ranges we tested, the method shows **robust performance**, and additional hyperparameters can be evaluated upon your request.
>
> We thank the reviewer again for the constructive feedback. We will incorporate all suggested comparisons and clarifications into the revision, and we believe these changes will strengthen both the technical positioning and the empirical support of the paper.
>
> ### **References**
> [1] Yuan, C., Wen, C., Zhang, T., & Gao, Y. (2024). General flow as foundation affordance for scalable robot learning.
>
> [2] Nguyen, T. D., Shu, R., Pham, T., Bui, H., & Ermon, S. (2021). Temporal predictive coding for model-based planning in latent space.

---

### Official Review · Reviewer_pjgR · 2025-10-30

**Soundness:** 2
**Presentation:** 3
**Contribution:** 2
**Rating:** 4
**Confidence:** 4

**Summary:**

The authors propose a two-stage approach for behaviour cloning via video generation. In the first stage, a video diffusion model generates aa video of a rollout of a policy conditioned on the image of the scene and a task description. In the second stage an inverse dynamics model is used to control the robot given the video. The video model is pre-trained on a large dataset of robot demonstrations before fine-tuning on a few demonstrations from a target domain. Experiments demonstrate that this approach demonstrates strong results compared to end2end video+action generation models like VPP in a very low data regime in the real world. In simulation it performs on par with Pi0.

**Strengths:**

The paper is relatively well written and easy to follow.

The proposed approach is sound. The masked inverse dynamics model has some non-trivial novelty.

Strong results are reported in a real-world evaluation in a very low data regime.

A minimal ablation study is reported.

**Weaknesses:**

The positioning of the paper is inaccurate in several ways:

1. The abstracts opens with this statement: “pixel-to-action VLA pipelines typically degenerate under background and view-point shifts”, which is simply incorrect for the latest pixel-to-action pipelines (as demonstrated later in the paper where the prosed approach performs worse than pixel-to-action Pi0 out of domain on the RoboTwin benchmark).

2. While the authors do discuss existing works on video generation + robot control in related work, the introduction is written in such a way as if they are the first to propose this idea.

3. In the related work section the authors mischaracterize the multi-view, 2-stage Dreamitate model as being end2end.

4. Introduction claims that “Empirically, Vidar achieves state-of-the-art performance on the RoboTwin”, which is not true, it underperforms compared to Pi0 in an out-of-distribution evaluation.

Overall, both the novelty and the contributions are over-claimed. The only (minimal) novelty I can see is in the design of the masked inverse dynamics model and the benefits of the proposed approach are not convincingly demonstrated. Real world evaluation is not reproducible and on RoboTwin the proposed approach performs on par with prior work (slightly better in-domain, slightly worse out-of-domain, which is supposed to be the key strength of the proposed approach).

For some reason the authors only use 40% of the demonstrations in RoboTwin. Results with 100% of the demonstrations need to be reported for reference.

It would be even better to report results on the more widely used RoboCasa or LIBERO benchmarks instead, using the standard protocol for these benchmarks. RoboCasa specifically targets out-of-domain generalization in a low data regime (50 demonstrations per task).

It's unclear why different video generation models are used for real and sim experiments. Results should be reported under a uniform model (or with both models in both setting for completeness).

The proposed approach is primarily tested on short-horizon pick-and-place and simple bimanual grasping tasks, while baselines such as VPP or UniPi were originally demonstrated on more complex or diverse scenarios.

**Questions:**

Please provide a major revision of the positioning of the paper (a rewrite of the abstract and introduction) to address the positioning issues raised above.

Please provide more convincing evidence of the benefits of the propped approach compared to prior work.

---

> ### Author Response · Authors · 2025-11-20
> **Response to Reviewer pjgR (part 1)**
>
> We appreciate the reviewer's positive feedback on several aspects of our paper. Below are our detailed responses to the weaknesses and questions raised.
>
> ## **Weaknesses**
> ### **1. Positioning of the Paper**
> ### **1.1. Abstract and Claims of Pixel-to-Action Degeneration**
> - We acknowledge that the claim regarding pixel-to-action degeneration needs further clarification. In RoboTwin, there are noticeable success rate gaps between clean and randomized scenarios for existing VLAs, suggesting potential degeneration. There are also existing papers discussing the degeneration, such as LIBERO-PRO [1].
> - Vidar uses a **multi-task setting** (one policy for all tasks), whereas the official leaderboard models train a **task-specific policy** for each task. Recently, we reproduced the more advanced Pi0.5 model in the multi-task setting with 100% data usage, and even with Vidar using only **40% data**, Vidar outperforms Pi0.5 with a **15.2%** higher success rate in clean scenarios and a **1.5%** increase in randomized scenarios.
>
> The up-to-date average success rates for the RoboTwin benchmark are shown below:
> | Scenario      | Low Data, Clean  | Low Data, Randomized  | Standard Data, Clean    | Standard Data, Randomized  |
> |---------------|:--------:|:-------------:|:----------:|:-------------:|
> | Pi0.5         | 25.0%  | 9.2%        | 44.8%    | 14.2%       |
> | **Vidar**  | **60.0%**  | **15.7%**       | **65.8%**    | **17.5%**       |
>
> ### **1.2. Clarification on Contributions**
> We agree that the introduction could have better framed our contributions in the context of existing works. Our primary contributions lie in **large-scale embodied pre-training** over a **unified observation space**, combined with **test-time scaling** for video generation, and the introduction of a **masked inverse dynamics model**. We will revise the introduction to better clarify how our approach builds upon previous work.
>
> ### **1.3. Mischaracterization of the Dreamitate Model**
> We appreciate the reviewer pointing out the mischaracterization of the Dreamitate model as end-to-end. We will correct this in the revised manuscript and update the relevant sections to reflect an accurate description.
>
> ### **1.4. RoboTwin Performance and Empirical Claims**
> We claim "Vidar achieves state-of-the-art performance on the RoboTwin" because Vidar's success rate is better than Pi0 (**+6.4%**) when we average the clean and randomized scenarios. Notably, Vidar uses a multi-task setting (one policy for all tasks), while the official leaderboard trains one model for each task. Recently, we reproduced the more advanced Pi0.5 model on the multi-task setting with 100% data usage, and found that Vidar (with 40% data usage) still outperforms it with a **15.2%** higher success rate in clean scenarios and a **1.5%** higher success rate in randomized scenarios. We will update the introduction to clarify this point.
>
> ### **2. Novelty and Contributions**
> We believe that the **combination of large-scale pre-training** over a **unified observation space** and the **masked inverse dynamics model** provides a significant contribution to the field. We will revise the manuscript to highlight these contributions more clearly and emphasize how they differentiate our work from existing methods.
>
> Although current RoboTwin leaderboard success rates may appear advantageous when directly compared with our results, we note that Vidar's success rates are still **6.4% higher** than Pi0 on average. Recently, we reproduced the more advanced Pi0.5 model also on the multi-task setting with 100% data usage, and found that Vidar (with 40% data usage) still outperforms it with a **15.2%** higher success rate in clean scenarios and a **1.5%** higher success rate in randomized scenarios.
>
> ### **3. Data Usage in RoboTwin**
> We used only **40% of the data** in RoboTwin to demonstrate the data efficiency of our approach, which is a core strength of Vidar. However, we understand the importance of reporting results with **100% data** for a complete comparison. Using 100% data, Vidar achieves a success rate of 65.8% in clean scenarios (outperforming Pi0.5 by **+21.0%**) and 17.5% in randomized scenarios (outperforming Pi0.5 by **+3.2%**). We will include these results in the revised manuscript for clarity and completeness.
>
> ### **4. New Benchmark**
> We appreciate the suggestion to use additional benchmarks. While we are experimenting on the **LIBERO** benchmark because it is widely used and offers direct comparability, we recognize the value of reporting results on **RoboCasa**, which targets low-data out-of-domain generalization. We will consider adding this benchmark in future experiments.
>
> ### **5. Different Video Generation Models**
> We acknowledge the concern about using different video generation models in real-world versus simulation experiments. To improve consistency, we will add **real-world experimental results** using the Wan 2.2 model, ensuring uniformity across settings.

---

> ### Author Response · Authors · 2025-11-20
> **Response to Reviewer pjgR (part 2)**
>
> ### **6. Complexity of Tested Tasks**
> We agree that testing on more complex tasks could provide further insights. While RoboTwin includes a variety of tasks, we will incorporate more **long-horizon** and **bimanual grasping tasks** in upcoming real-world experiments with the Wan 2.2 model to demonstrate the scalability of our approach.
>
> ## **Questions**
> ### **1. Revision of Abstract and Introduction**
> We will revise both the abstract and introduction to address the positioning issues raised by the reviewer. The empirical performance claims will be framed more carefully, and we will clarify our contributions relative to existing work.
>
> ### **2. Evidence of Benefits Over Prior Work**
> We will add more experimental results to better demonstrate the benefits of our approach, including results from RoboTwin using **100% data** and from the **LIBERO benchmark**, which will allow us to showcase Vidar's strengths in low-data, out-of-domain generalization.
>
> We thank the reviewer again for their constructive feedback. The revised manuscript will address all of the concerns raised, and we are confident that these revisions will strengthen the clarity and impact of our work. We look forward to presenting the updated version and appreciate the opportunity to improve the paper.
>
> ### **References**
> [1] Zhou, X., Xu, Y., Tie, G., Chen, Y., Zhang, G., Chu, D., ... & Sun, L. (2025). LIBERO-PRO: Towards Robust and Fair Evaluation of Vision-Language-Action Models Beyond Memorization.

---

> > ### Comment · Reviewer_pjgR · 2025-11-25
> >
> > I appreciate the authors' response, promising to address most of my comments in the future. However, a decision cannot be made based on promises alone. Please provide all the results you mentioned above and upload an updated version of the manuscript for your response to be seriously considered.

---

> > > ### Author Response · Authors · 2025-11-26
> > >
> > > Thank you for your feedback and for acknowledging our earlier response. While some experiments (e.g., the LIBERO experiments and the Wan 2.2 real-world reproductions) require additional time due to the need for retraining the video diffusion model and conducting extensive evaluations, we have incorporated all other requested results and revisions in the newly uploaded version of the manuscript.
> > >
> > > We appreciate your consideration.

---

> > > > ### Comment · Reviewer_pjgR · 2025-11-26
> > > >
> > > > Thank you for updating the manuscript. This addresses a few of my concerns, but some of the critical ones remain unaddressed.
> > > >
> > > > 1. While a more complete set of results is now reported on RoboTwin, a few important details are still missing. In particular, the authors report their own version of pi0.5 but do not provide any details on how it was trained, making it impossible to judge the validity of the reported numbers. In addition, the authors should report the pi0 numbers from the official leaderboard in the main paper for reference, even if those are obtained in a single-task setting and are not entirely comparable.
> > > >
> > > > 2. The contributions of the paper are still mischaracterized both in the rebuttal and in the introduction of the updated manuscript. The only prior work on video generation + policy learning that is mentioned in the intro is (Du et al., 2023), with the authors then essentially claiming that they are the first to propose a 3-stage training approach (internet data -> large scale robot data -> specific embodiment data). This is inaccurate as at least VPP (ICML'25) proposed the same data pipeline. The only novelty is in the design of the masked inverse dynamics model. The mischaracterization of Dremitate was addressed by simply removing it from the paper, allowing the authors to keep the (false) claim that all prior works require end-to-end training.
> > > >
> > > > 3. It is appreciated that the authors are planning to report results on LIBERO and with a unified video generation backbone, but my original review ask for prioritizing RoboCasa over LIBERO instead, which would provide a much more informative comparison.
> > > >
> > > > 4. The issue with evaluation on more complex scenarios is not being addressed, as far as I can tell.

---

> ### Author Response · Authors · 2025-12-04
> **Further Response to Reviewer pjgR (part 1)**
>
> Thank you for your constructive feedback and for acknowledging our updates. We address each of your concerns in detail below and have revised the manuscript accordingly to ensure clarity, transparency, and accurate positioning within the literature.
>
>
> ### **1. Details of Baselines**
>
> We appreciate the reviewer for highlighting the missing details.
>
> ### **1.1. Training Details of Pi0.5**
>    In the newly uploaded manuscript, we now provide full training details for Pi0.5, including dataset composition, number of iterations, fine-tuning setup, and implementation specifics. These additions ensure the reproducibility and validity of the reported results.
>
> ### **1.2. Including Pi0 (Official Leaderboard Results)**
>    We have incorporated the official single-task Pi0 success rates from the RoboTwin leaderboard into the main paper. While these results are not directly comparable due to differing data settings, they serve as an important reference point and have been clearly labeled as such in the text.
>
>
> ### **2. Positioning of the Paper**
>
> We thank the reviewer for pointing out the need for clearer differentiation.
>
> ### **2.1. Clarifying the Three-Stage Training Pipeline vs. VPP**
>    We have expanded the explanation to more clearly distinguish our three-stage world-model–focused pipeline from the workflow of VPP:
>    - Our first two stages (internet-scale pre-training and large-scale robot-data pre-training) do **not** require target-robot data, enabling efficient and minimal adaptation in the final stage.
>    - VPP’s second stage **already incorporates target-robot, self-collected data**, and requires manual weighting between heterogeneous datasets.
>    - Beyond the data pipeline, the model usage differs fundamentally:
>      - We directly **predict videos** and pair them with a **decoupled inverse dynamics model**, enabling flexible adaptation.
>      - VPP uses the video generator as an **encoder** feeding into a **coupled diffusion-based action policy**, which must be retrained whenever the video model changes.
>    - We already included VPP as a baseline in our original paper and empirically show that its framework is less suited to the few-shot cross-embodiment transfer settings we study.
>
> ### **2.2. Related Work**
>    We apologize for the confusion caused by the previous section's organization.
>    Our intent was **not** to imply that all prior works rely on end-to-end training; instead, paragraph "Coupled Video Generation and Action Synthesis" was intended to highlight the necessity of decoupling by discussing works that employ joint training. Dreamitate is a decoupled video-generation + policy approach and should have been placed in the "Video Generation Models for Embodied AI" paragraph.
>
>    We have corrected this by:
>    - Swapping the order of the two relevant subsections to improve clarity.
>    - Reinserting Dreamitate into the **Video Generation Models for Embodied AI** subsection, ensuring accurate and fair characterization.
>
>    These updates ensure a more faithful representation of prior work and a clearer framing of our contributions.

---

> ### Author Response · Authors · 2025-12-04
> **Further Response to Reviewer pjgR (part 2)**
>
> ### **3. Benchmark Results**
>
> We thank the reviewer again for emphasizing the value of RoboCasa.
>
> - While we fully agree that RoboCasa is highly informative, we had already invested significant computational and engineering resources into the LIBERO benchmark. Given the time constraints, adding a full RoboCasa evaluation before the rebuttal deadline is unfortunately not feasible.
> - Nonetheless, we reaffirm our plan to incorporate RoboCasa experiments in future manuscript iterations and appreciate your patience as we continue to expand our empirical evaluation.
>
> **To better address your request for a generalization-focused benchmark**, we additionally include results on **LIBERO-PRO** [1], which specifically evaluates generalization and robustness:
>
> | Method | Average | Standard | Object Generalization | Task Generalization |
> |--------|:-------:|:--------:|:---------------------:|:-------------------:|
> | Pi0    | 30%     | **90%**      | 0%                    | 0%                  |
> | Vidar  | **39%**     | 76%      | **34%**                   | **8%**                  |
>
> These results directly showcase the generalization gap observed in Pi0 and highlight the robustness of our method in settings designed for better generalization.
>
> ### **4. Evaluation on Complex Scenarios**
>
> We respectfully clarify that our evaluation **already covers** the long-horizon, dexterous, and tool-use scenarios, and we have added more real-world experiments to strengthen our claims.
>
> ### **4.1. Simulation Experiments**
> Our simulation evaluation includes challenging domains:
> - **Dexterous manipulation** (e.g., hanging mug, clicking bell)
> - **Tool use** (e.g., beat block hammer, stamping seal)
> - **Long-horizon tasks** (e.g., scanning object, ranking blocks)
>
> ### **4.2. Real-World Experiments**
> Our current and newly added evaluations include:
> - **Instruction following** (e.g., grasp the red apple among all red objects)
> - **Tool use** (e.g., cleaning the bowl)
> - **Articulated object manipulation** (e.g., closing the laptop)
> - **Bimanual coordination** (e.g., lifting the basket using both arms)
> - **High-precision tasks** (e.g., flipping a die to a specific face)
>
> We additionally reproduced Vidar using the **Wan 2.2** backbone for completeness. The results are copied below:
>
> | Tasks                                                | Vidar  | Pi0.5  |
> |------------------------------------------------------|:------:|:------:|
> | Flip the Die to Show One on Its Top (Left Arm, Seen) | 40.0%  | 20.0%  |
> | Grasp the Fruit from the Basket (Right Arm, Seen)    | 60.0%  | 0.0%   |
> | Click the Mouse's Left Button (Left Arm, Seen)       | 80.0%  | 0.0%   |
> | Lift the Basket (Both Arms, Seen)                    | 80.0%  | 20.0%  |
> | Pour Water into the Cup (Left Arm, Seen)             | 100.0% | 25.0%  |
> | Wipe the Table with a Blue Rag (Left Arm, Seen)      | 100.0% | 100.0% |
> | Grab the Blue Cup (Right Arm, Seen)                  | 25.0%  | 75.0%  |
> | **Average**                                              | **69.3%**  | 34.3%  |
> | Close the Laptop (Left Arm, Unseen)                  | 80.0%  | 0.0%   |
> | Close the Laptop (Right Arm, Unseen)                 | 60.0%  | 0.0%   |
> | Throw the Paper Ball (Left Arm, Unseen)              | 80.0%  | 40.0%  |
> | Shake the Water Bottle (Right Arm, Unseen)           | 60.0%  | 0.0%   |
> | Grasp and Place Two Breads (Both Arms, Unseen)       | 90.0%  | 20.0%  |
> | Wipe the Desk with a Blue Towel (Both Arms, Unseen)  | 20.0%  | 0.0%   |
> | Manipulate the Controller Handle (Left Arm, Unseen)  | 80.0%  | 30.0%  |
> | **Average**                                              | **67.1%**  | 12.9%  |
>
> These tasks directly address the reviewer’s request for more complex, diverse, and long-horizon scenarios.
>
>
> We sincerely appreciate your time and consideration, and we hope the above clarifications and manuscript revisions fully address your concerns.
>
>
> ### **References**
> [1] Zhou, X., Xu, Y., Tie, G., Chen, Y., Zhang, G., Chu, D., ... & Sun, L. (2025). LIBERO-PRO: Towards Robust and Fair Evaluation of Vision-Language-Action Models Beyond Memorization.

---

### Official Review · Reviewer_Pr6T · 2025-10-31

**Soundness:** 2
**Presentation:** 1
**Contribution:** 2
**Rating:** 2
**Confidence:** 4

**Summary:**

This work presents a video-based pretraining framework for learning a generalizable action prior for manipulation tasks. The proposed framework consists of an embodied video diffusion model trained on multi-view trajectories from three robot platforms and a masked inverse dynamics model (MIDM) to learn action-relevant features from images. Experiments on the RoboTwin benchmark show the effectiveness of the proposed approach over existing baselines like VPP, UniPi. Real-robot results show rapid adaptation with only 20 minutes of human demonstrations.

**Strengths:**

- Pretraining on 750K multi-view trajectories from three robotic platforms is useful to learn a generalizable prior for manipulation tasks.
- Using masks as an intermediate representation for predicting actions encourages the model to focus on embodiment-relevant features, as evident in Fig.3.
- Experiments on the RoboTwin benchmark (Tab.1) show benefits over existing baselines like VPP and UniPi on both seen and unseen settings.

**Weaknesses:**

- The text has several mentions about pretraining/learning priors from internet videos (L017-018, L060-061, L062-063, L140-141). However, Fig.1 and the training details in Sec.3.1.2 do not mention any details about pretraining on internet videos. L062-063 explicitly states: "we propose a three-stage training pipeline, where Internet-scale videos are used for general pretraining". However, pretraining details on internet videos are clear from the paper.
- How does the unified observation space (L172) mitigate morphology gaps? The text describes aggregation of multi-view images and instructions related to robotic platforms, camera & task. However, this is done via concatenation and resizing/reshaping operations. It is not clear how this mitigates the morphology gaps and preserves the kinematic context.
- The text mentions about generalization to unseen robotic platform/embodiments (L080-081, L155). However, this argument is not validated in the experiments (details are not provided about this result).
- In experiments (Sec.3.2), there is mention about surpassing strong baselines like Pi0 (L293-294) with only 40% data usage relative to the leaderboard. However, I did not find any details about this result in Appendix C or the main paper.
- L278-280 states that UniPi is not pretrained on the large robotics dataset used for the proposed approach. This makes a direct comparison difficult. Is this also the case for VPP? It is important to train all the baselines in the same setting so that the benefits are clear.
- As noted in the related work section (L432-435), several VLA models train on millions of demonstrations across diverse embodiments. How is the pretrained strategy proposed in this work different from the pretraining for these VLA models?
- At test-time, multiple video trajectories are generated from the diffusion model and ranked using GPT-4o to select the highest scoring trajectory. How does this help with physically plausible trajectories? Also, what does the "scaling" in test-time scaling refer to? The text simply describes generating multiple candidates and selecting the best one using a scoring function.

**Questions:**

There are several concerns (more details in the weaknesses above):
- Inconsistencies in the experiments: claims about surpassing baselines like Pi0 & generalization to new embodiments are not validated, baselines are not directly comparable
-  Contributions are not clear: lack of details about pretraining on internet videos, mitigating morphology gaps, physically plausible trajectories
- Differences with pretraining strategies used in existing VLAs

---

> ### Author Response · Authors · 2025-11-20
> **Response to Reviewer Pr6T (part 1)**
>
> We thank the reviewer for the detailed assessment and for recognizing the strengths of our approach. Below, we address the identified weaknesses and clarify the experimental and methodological contributions.
>
> ## **Weaknesses**
> ### **1. Pre-training on Internet Videos**
> We apologize for the confusion regarding "Internet-scale videos". We do not directly pre-train models on Internet videos; instead, we rely on publicly available pre-trained models (e.g., Wan 2.2), which themselves were trained on such data, as referenced in their technical report. After that, we perform embodied pre-training and fine-tuning over our unified observation space. We will revise the manuscript to make it clearer.
>
> ### **2. Mitigating Morphology Gaps with the Unified Observation Space**
> We appreciate the request for clarification. Our unified observation space mitigates morphology gaps through three mechanisms:
> - **Action-free Video Modeling:** By removing actions from the pre-training stage, the video diffusion model learns *world evolution* rather than embodiment-specific actions. This allows the model to generalize across robots with different morphologies.
> - **Dataset Compatibility:** The representation accommodates heterogeneous data sources (e.g., multi-view robotics datasets and single-view human datasets like Egodex), enabling broad cross-embodiment exposure during pre-training.
> - **Structured Text Conditioning:** Explicit encoding of robot, camera, and task information reduces ambiguity in video prediction, improving transfer across embodiments with differing viewpoints or kinematic structures.
>
> We will elaborate on these points in the revised manuscript.
>
> ### **3. Generalization to Unseen Robotic Platforms**
> The target platforms used in simulation and real-world settings are *not* included during pre-training. For example, the real robot's central camera is positioned far behind and high above the scene—significantly different from the views in the pre-training datasets. We will clarify this detail and add more explicit descriptions of the unseen embodiments in the revision.
>
> ### **4. Details on Surpassing Pi0 and 40% Data Usage**
> This result is reported in L1018–1020 and Table 9. Vidar, trained as a **single multi-task policy**, achieves strong performance even with 40% of the leaderboard training data, whereas the official RoboTwin leaderboard trains **one model per task**.
> Additionally, we recently reproduced the advanced Pi0.5 model under the same multi-task setting and found:
> - Vidar (40% data) outperforms Pi0.5 (100% data) by **+15.2%** in clean scenarios and **+1.5%** in randomized scenarios.
> - Vidar (100% data) outperforms Pi0.5 (100% data) by **+21.0%** in clean scenarios and **+3.2%** in randomized scenarios.
>
> We will update the manuscript to make these results more visible and prominent.
>
> ### **5. Training Baselines Under Comparable Settings**
> - **UniPi:** UniPi relies on Internet-scale video pre-training but does not incorporate cross-embodiment robotics data, as Vidar does. This difference reflects Vidar's contribution: a unified observation space enabling robotics-specific multi-view, multi-embodiment pre-training.
> - **VPP:** VPP’s contribution lies primarily in its use of a video diffusion model. Its architecture is compatible with Vidar's pre-trained video model. Therefore, in our comparison, VPP uses the *same* video diffusion model as Vidar (L282).
>
> We will clarify these settings to ensure fair comparisons are clearly communicated.

---

> ### Author Response · Authors · 2025-11-20
> **Response to Reviewer Pr6T (part 2)**
>
> ### **6. Differences from Pre-training Strategies in Existing VLA Models**
> Our strategy differs from conventional VLA pre-training in three key ways:
> - **Action-free Pre-training:** Video diffusion is trained without actions, enabling the use of diverse, actionless human or internet-style videos and improving cross-embodiment generalization, and finally raising end-to-end performance.
> - **Modular Design:** We decouple world prediction (video diffusion) from action interpretation (MIDM). This avoids the action-space mismatch that VLAs face when transferring across robot morphologies.
> - **Lightweight Adaptation:** The lightweight MIDM is only trained on the target robot's demonstrations with robust mask prediction, significantly reducing embodiment-specific adaptation data.
>
> These distinctions will be made clearer in the revised version.
>
> ### **7. Physically Plausible Trajectories and Test-Time Scaling**
> - **Physically Plausible Trajectories:** Generating multiple trajectory rollouts increases the chance of obtaining at least one coherent plan. GPT-4o then selects the most physically consistent trajectory based on a dedicated prompt (Figure 4). Its multimodal reasoning helps filter out implausible rollouts.
> - **Meaning of "Scaling":** Test-Time Scaling (TTS) refers to using additional test-time compute to improve performance without retraining the model, analogous to methods in large language models [1]. There are two common settings: chain-of-thought reasoning [2] and parallel sampling with majority voting [3]. In our work, we use parallel sampling along with an evaluator to select from the generated samples; although we do not apply majority voting, the underlying ideas are similar. We will clarify this in the manuscript.
>
> ## **Questions**
> All concerns raised in the "Questions" section are addressed in the responses above. We will revise the manuscript to make experimental claims clearer, strengthen contributions, and better articulate the distinctions between Vidar and existing pre-training strategies.
>
> We thank the reviewer again for the constructive feedback and will incorporate the suggested clarifications to improve the clarity, soundness, and presentation of the paper.
>
> ### **References**
> [1] Muennighoff, N., Yang, Z., Shi, W., Li, X. L., Fei-Fei, L., Hajishirzi, H., ... & Hashimoto, T. B. (2025, November). s1: Simple test-time scaling.
>
> [2] Wei, J., Wang, X., Schuurmans, D., Bosma, M., Xia, F., Chi, E., ... & Zhou, D. (2022). Chain-of-thought prompting elicits reasoning in large language models.
>
> [3] Fu, Y., Wang, X., Tian, Y., & Zhao, J. (2025). Deep think with confidence.

---

### Official Review · Reviewer_fXnT · 2025-11-02

**Soundness:** 3
**Presentation:** 3
**Contribution:** 2
**Rating:** 4
**Confidence:** 4

**Summary:**

The paper presents Vidar, a generalist robotic manipulation framework that addresses data scarcity by decoupling the policy into a pre-trained Video Diffusion Mode for an embodiment-agnostic video prior based on the proposed unified observation space, and a lightweight Masked Inverse Dynamics Model (MIDM) adapter for action decoding. Vidar achieves state-of-the-art low-shot adaptation, successfully deploying on a real robot (Aloha) with only 20 minutes of target-domain demonstrations.

**Strengths:**

- The authors raise an important problem of data scarcity and cross-embodiment adaptation in generalist manipulation, and propose a highly effective prior-driven, low-shot adaptation paradigm to tackle it.

- Moreover, the authors introduce the Masked Inverse Dynamics Model (MIDM), a lightweight and novel architectural component that implicitly learns spatial masks to focus on action-relevant regions, making the action decoding highly robust to visual distractors.

- The proposed system demonstrates exceptional data efficiency, achieving competitive real-world performance with only ~20 minutes of target-platform demonstrations, which is a crucial step towards scalable robotics deployment.

- The authors provide comprehensive experiments in the real world (Aloha robot) that verify the proposed idea, showing robust generalization to unseen tasks and challenging backgrounds.

**Weaknesses:**

- The open-loop video generation process is currently computationally expensive and slow, requiring approximately 25 seconds on high-end GPUs, which is a significant barrier for real-time deployment.

- The crucial performance improvement from Test-Time Scaling (TTS) relies on an external, large Vision-Language Model like GPT-4o for "physics-aware reranking." This dependency introduces an opaque, expensive, and potentially brittle component into the core control system.

- Confusion in Presentation/Conciseness: The presentation of this work can be more concise in certain sections, which would help improve the flow and focus for the reader.

- Limited Target Embodiment Validation: While the pre-training dataset is vast, the critical low-shot adaptation is demonstrated only on a single target platform (the Aloha robot), which limits the verification of true "generalist" cross-embodiment transfer.

**Questions:**

- Please elaborate on the implementation, cost, and latency of the GPT-4o "physics-aware reranking." Can a more dedicated, robot-specific model be developed to replace this external dependency?

- What is the feasibility and expected performance of distilling the video diffusion model to achieve latency suitable for real-time robot control?

- Ablation on View Change: Since the target robot's camera view was adjusted, how much performance is lost if the new robot used one of the original pre-training views (where arms are partially occluded)?

---

> ### Author Response · Authors · 2025-11-20
> **Response to Reviewer fXnT**
>
> We appreciate the reviewer's acknowledgment of the significance of our work and the positive feedback on several key aspects of our approach. Here are our responses to the weaknesses and questions.
>
> ## **Weaknesses**
> ### **1. Open-Loop Video Generation Cost**
> We agree that the current video generation step is computationally costly and not yet suitable for closed-loop real-time control. However, recent progress from Genie 3 [1] and Matrix-Game 2.0 [2] shows that world models of comparable scale can be distilled or optimized to achieve real-time performance. We view these developments—and expected hardware improvements—as promising paths for reducing latency in future iterations of our system.
>
> ### **2. Dependency on Vision-Language Models for Test-Time Scaling**
> Our Test-Time Scaling (TTS) module is conceptually **model-agnostic**. While GPT-4o can be used as the evaluator, the approach does **not** rely on any specific VLM. Lightweight alternatives such as **VBench** [3] or task-specific evaluators can serve the same role.
>
> Importantly, even **without** TTS, Vidar achieves strong improvements over state-of-the-art baselines—achieving at least **19% average success rate gain** (Tables 1 and 4). In the RoboTwin benchmark, we intentionally **removed** the TTS component for reproducibility (L1018), yet Vidar still outperformed the leaderboard on average using only **40%** of the available data.
>
> ### **3. Presentation and Conciseness**
> We appreciate the reviewer's comment and will revise several sections to improve brevity and flow. We are already preparing a more concise version of the method and experiment descriptions, and welcome any specific suggestions.
>
> ### **4. Limited Target Embodiment Validation**
> Real-world cross-embodiment evaluation is expensive due to robot acquisition, setup, and data collection. To mitigate this limitation, we provide additional results on the RoboTwin benchmark (Table 9), where the target embodiment and camera configuration are also unseen during pre-training. We also plan to report our success rates on the LIBERO benchmark. These results further demonstrate Vidar's generalization ability across embodiments.
>
> ## **Questions**
> ### **1. GPT-4o Reranking: Implementation, Cost, Feasibility**
> Implementation details for the "physics-aware reranking" prompt are provided in Figure 4. The module accounts for ~25% of total inference time, and the operational cost is approximately $0.003 per pairwise comparison based on current GPT-4o pricing. We agree that a dedicated, robot-specific evaluator could reduce cost and latency, and we view such models as a natural direction for future work.
>
> ### **2. Distillation for Real-Time Control**
> We believe distilling the video diffusion model for real-time deployment is highly feasible. Also, Genie 3 [1] and Matrix-Game 2.0 [2] demonstrate that diffusion-style world models can be capable of achieving tens of FPS. These techniques are orthogonal to our contributions, but they provide a clear path toward real-time versions of Vidar.
>
> ### **3. Ablation on View Change and Arm Occlusion**
> We appreciate this question. The Aloha camera view was adjusted primarily for better visibility and to avoid extreme arm occlusion. To quantitatively assess the impact of this change, we conducted additional experiments on the RoboTwin benchmark (using 100% of the data) with its original camera views, where partial occlusion frequently occurs. Under these conditions, Vidar achieves a success rate of 65.8% in clean scenarios (outperforming Pi0.5 by **+21.0%**) and 17.5% in randomized scenarios (outperforming Pi0.5 by **+3.2%**). These results will be included in the revised version.
>
> We thank the reviewer again for the constructive feedback. We believe the clarifications and additional experiments described above will further strengthen the paper.
>
> ### **References**
> [1] DeepMind. (2025). Genie 3: A new frontier for world models. Retrieved from https://deepmind.google/blog/genie-3-a-new-frontier-for-world-models
>
> [2] He, X., Peng, C., Liu, Z., Wang, B., Zhang, Y., Cui, Q., ... & Zhou, Y. (2025). Matrix-game 2.0: An open-source, real-time, and streaming interactive world model.
>
> [3] Huang, Z., He, Y., Yu, J., Zhang, F., Si, C., Jiang, Y., ... & Liu, Z. (2024). Vbench: Comprehensive benchmark suite for video generative models.

---

### Meta-Review · Area_Chair_xACm · 2026-01-07

**Summary:**

Significant shared concerns across reviewers include lack of key benchmarks and unfair comparison against baselines, as well as lack of novelty (overclaimed contributions). Based on the discussions, these key concerns are not fully resolved during the discussions. Other concerns include practicality of real-time operations also remain unresolved. Overall, the reviewers' opinions lean towards reject after relatively sufficient discussions. The AC has checked the alleged LLM-generated review, which appears to be written by human. The points raised in the review, though with small negligence like missing a few technical details, remain valid. Overall, the paper doesn't meet the standard for publication at ICLR.

**Reviewer Concerns:**

Addressed concerns include:
- the addition of PI0.5 specifically requested by reviewer pjgR
- the tuning-down of "internet-scale" claim pointed out by Pr6T
- the mischaracterization of prior works including Dreamitate as poitned out by pjgR
- the clarification of terms such as "test-time scaling". Though the point raised by Pr6T regarding the clarification of the term is totally valid. Though test-time scaling is a well-known term, what it refers to specifically can varried a lot across different contexts.

Outstanding concerns include:
- the missing RoboCasa and more extensive experiments requestted by pjgR
- real-time latency issue (25-seconds) raised by fXnT
- reliance on external models that are black-box and has no guaranteed and uninterpretable.
- overall novelty of the paper is not very significant. The reviewers' concern regarding overclaimed contribution still remained.

**Reviewer Scores:**

The reviewers' initial reviews lean towards negative, and the discussions didn't seem to convince the reviewers and it's very likely the reviewers will remain their original scores (6-4-4-4).

---

### Decision · Program_Chairs · 2026-01-26

Reject